# epDevAtlas: mapping GABAergic cells and microglia in the early postnatal mouse brain

Josephine K. Liwang[1,4], Fae N. Kronman [1], Hyun-Jae Pi[1], Yuan-Ting Wu[1], Daniel J. Vanselow [1], Steffy B. Manjila [1,5], Deniz Parmaksiz [1,2], Donghui Shin [1,6], Yoav Ben-Simon[3], Michael Taormina [3], Sharon W. Way [3], Hongkui Zeng [3], Bosiljka Tasic [3], Lydia Ng[3] & Yongsoo Kim [1,2] ✉

During development, brain regions follow encoded growth trajectories. Compared to classical brain growth charts, high-definition growth charts could quantify regional volumetric growth and constituent cell types, improving our understanding of typical and pathological brain development. Here, we create high-resolution 3D atlases of the early postnatal mouse brain, using Allen CCFv3 anatomical labels, at postnatal days (P) 4, 6, 8, 10, 12, and 14, and determine the volumetric growth of different brain regions. We utilize 11 different cell type-specific transgenic animals to validate and refine anatomical labels. Moreover, we reveal region-specific density changes in γ-aminobutyric acid-producing (GABAergic) neurons, cortical layer-specific cell types, and microglia as key players in shaping early postnatal brain development. We find contrasting changes in GABAergic neuronal densities between cortical and striatal areas, stabilizing at P12. Moreover, somatostatin-expressing and vasoactive intestinal peptide-expressing cortical interneurons undergo regionally distinct density changes. Remarkably, microglia transition from high density in white matter tracks to gray matter at P10, and show selective density increases in sensory processing areas that correlate with the emergence of individual sensory modalities. Lastly, we create an open-access web-visualization (https://kimlab.io/brain-map/epDevAtlas) for cell-type growth charts and developmental atlases for all postnatal time points.

Brain growth charts provide quantitative descriptions of brain development, primarily focusing on changes in macroscopic brain volume and shape analysis[1,2]. However, enhanced brain growth charts that track individual brain regions and cell types could significantly improve our understanding of normal brain development, as well as pathological deviations associated with various neurodevelopmental disorders. In rodents, the first two postnatal weeks of brain development are broadly equivalent to human brain developmental stages between late gestation and early childhood[3,4]. This period is characterized by a series of well-orchestrated and diverse events, including the generation and migration of neurons and non-neuronal cells, programmed cell death, and formation of synapses within a rapidly expanding brain volume[5–8]. Specifically, γ-Aminobutyric acid-producing (GABAergic) cell development plays a critical role in establishing the balance between excitatory and inhibitory networks, in coordination with glutamatergic cells[9–11]. For instance, cortical GABAergic neurons, born in the ganglionic eminences of the embryonic brain, undergo activity-dependent programmed cell death

[1]Department of Neuroscience and Experimental Therapeutics, College of Medicine, The Pennsylvania State University, Hershey, PA, USA. [2]Center for Neural Engineering, The Pennsylvania State University, State College, PA, USA. [3]Allen Institute for Brain Science, Seattle, WA, USA. [4]Present address: Princeton Neuroscience Institute, Princeton University, Princeton, NJ, USA. [5]Present address: Neuroscience Center of Excellence, School of Medicine, Louisiana State University Health Sciences Center, New Orleans, LA, USA. [6]Present address: Purdue University, West Lafayette, IN, USA. ✉e-mail: yuk17@psu.edu

in the early postnatal period to establish final densities expected in adulthood[12–16]. Microglia, the innate immune cells of the central nervous system (CNS), also have a critical role in brain development and wiring by facilitating GABAergic neurogenesis, neuronal migration, developmental neuronal apoptosis, synaptogenesis, and synaptic pruning[17–22]. Abnormalities in the developmental processes of these cell types have been implicated in various neurodevelopmental and psychiatric disorders[23–27]. Despite the significance of these brain cell types and emerging evidence of their regional heterogeneity[28–30], our knowledge of their quantitative changes in postnatally developing brains remains limited.

Recent advances in high-resolution 3D mapping methods with cell type specific labeling now allow for examination of the regionally distinct distribution of target cell types in the mouse brain[31–35]. Previously, we discovered that GABAergic neuronal subclasses exhibit highly heterogeneous density distributions across different regions, creating distinct cortical microcircuits in the adult mouse brain[31]. This regional heterogeneity may be partly attributed to their varying embryonic origins, birth dates, and programmed cell death. Indeed, cortical interneurons derived from both the medial (MGE) and caudal ganglionic eminence (CGE) undergo different rates of cell death[36]. Comparably, microglia exhibit both interregional and intraregional spatial density variations in the adult mouse brain, with higher densities observed in the hippocampus and basal ganglia[37,38]. However, it remains unclear how early these region-specific density patterns emerge, how different GABAergic cell subclasses undergo contrasting developmental changes, and how these changes occur in synchrony with microglial development to shape the mature brain cell type landscape.

One of the main challenges is the lack of 3D atlases for the developing mouse brain that integrate the spatiotemporal trajectories of brain cell types within a consistent spatial framework[39,40]. To address this need, a number of resources have been established, each with a unique focus. For instance, the developmental mouse brain common coordinate framework (DevCCF) offers a multi-modal high-resolution 3D reference map across embryonic and postnatal stages at select time points[40]. The Allen Developing Mouse Brain Atlas (ADMBA) emphasizes gene expression in situ hybridization (ISH) data across the same embryonic and postnatal stages[41]. Additionally, resources like the GenePaint Database and EMAP eMouse Atlas are valuable for their specific gene expression profiles and 3D developmental data[42,43], respectively. However, these resources still lack the fine temporal and spatial granularity crucial for capturing developmental brain growth.

For these reasons, we created a 3D early postnatal developmental mouse brain atlas (epDevAtlas) using serial two-photon tomography (STPT) imaging at postnatal days (P) 4, 6, 8, 10, 12 and 14, with anatomical labels based on the Allen Mouse Brain Common Coordinate Framework, Version 3.0 (Allen CCFv3)[44]. Additionally, we developed a pipeline to systematically register target cell types in epDevAtlas and to establish standard reference growth charts for GABAergic neurons, cortical layer-specific neurons, and microglia. Leveraging this new resource, we identified contrasting density changes of GABAergic neurons and microglia in cortical areas, as well as white-to-gray matter colonization of microglia during early postnatal periods. Equipped with web visualization, the 3D atlas and cell type density growth charts generated from this study provide a valuable suite of open science resources for understanding early postnatal brain development at cellular resolution and demonstrate the scalability of our approach for mapping other brain cell types.

## Results

### Creating 3D developmental brain atlases with CCFv3 anatomical labels

3D reference atlases provide essential spatial frameworks that enable registration and joint analysis of different brain data sources[39,44]. In this study, we created morphological and intensity averaged templates using early postnatal mouse brain samples acquired by high-resolution STPT imaging (Fig. 1a). Using Applied Normalization Tools (ANTs), individual samples at each age were iteratively averaged to create symmetrical templates with 20 μm-isotropic voxel size at postnatal days P4, 6, 8, 10, 12, and 14 (Fig. 1a; see Methods for more details). We then applied the Allen CCFv3 anatomical labels to our new templates by performing down registration of the P56 CCFv3 template to younger brain templates using ANTs, guided by manually marked major boundaries of distinct regions (e.g., midbrain-cerebellum) (Fig. 1b; see Methods for more details).

To validate and refine our registered CCFv3 labels at each age, we imaged brains from transgenic animals specifically selected for their differential cell type expression along previously defined anatomical borders (Table 1)[45,46]. Using individual or double recombinase driver lines crossed to appropriate reporter lines, we labeled specific cell populations in the brain (Table 1; Supplementary Tables 1–4). For cortical layers, for example, we used a *Slc32a1/Lamp5* intersectional mouse line for layer 1 (L1), *Calb2* for layers 2 and 3 (L2/3), *Nr5a1* for layer 4 (L4), *Rbp4* for layer 5 (L5), *Ntsr1* for layer 6a (L6a) and *Cplx3* for layer 6b (L6b) at P6, P10, and P14 (Fig. 1c). Subcortical expression and axonal projections from these and additional transgenic animals also helped to validate anatomical borders for other brain regions. For instance, cortico-thalamic projections detected in *Ntsr1* mouse brains specifically delineate thalamic regions (Fig. 1c). Moreover, *Gad2* mice help to delineate substructures of the striatum, including the caudo-putamen (CP) and external globus pallidus (GPe), which have markedly distinct *Gad2* expression in cells and passing fibers, respectively (Fig. 1d). Similarly, expression patterns from vasoactive intestinal peptide (*Vip*) mice mark the suprachiasmatic nucleus (SCH) (Fig. 1e), while in somatostatin (*Sst*) mice, the inferior colliculus (IC) is identified (Fig. 1f), and in *Pvalb* mice, the cerebellum (CB) is labeled (Fig. 1g). Together, these validations confirmed the accuracy of our anatomical labels across all early postnatal ages.

The resulting 3D epDevAtlas templates, with ontologically consistent anatomical labels, offer a unique opportunity to quantify detailed volumetric changes of various brain regions at different developmental time points (Source Data). Our data revealed a rapid expansion of brain volume, with approximately a two-fold increase in both whole brain and cerebral cortex averaged volumes between P4 and P14 (Fig. 1h). Moreover, the cerebellum showed the most substantial volumetric increase ( ~ four-fold) while diencephalic regions (i.e., thalamus, hypothalamus) exhibited the smallest volumetric increase (Fig. 1h). Comparisons of our STPT-based templates with DevCCF magnetic resonance imaging (MRI)-based templates at P4 and P14, and with a previously published MRI study[47], demonstrated minimal (less than 5%) volume changes with almost no morphological deformations in our STPT-based templates (Supplementary Fig. 1; Source Data).

### Developmental mapping of GABAergic neurons

To establish cell type growth charts during the first two postnatal weeks of mouse brain development, we built a computational workflow that detects genetically defined cell types and then maps their densities onto our newly developed epDevAtlas templates (Fig. 2a). The workflow involves high-resolution imaging data acquisition through STPT, machine learning (ML)-based cell detection using ilastik, image registration to age-matched epDevAtlas templates, and the generation of statistical outputs detailing signals and volumes per anatomical region-of-interest (ROI) (Fig. 2a)[33]. Our cell counting pipeline produces an organized data output of volume (mm³), counted cells, and cell densities (cells/mm³) for each anatomical brain region (Source Data). Additionally, this flexible pipeline can accommodate STPT data acquired at varying resolutions and has the potential for application with other high-resolution imaging data (e.g., light sheet

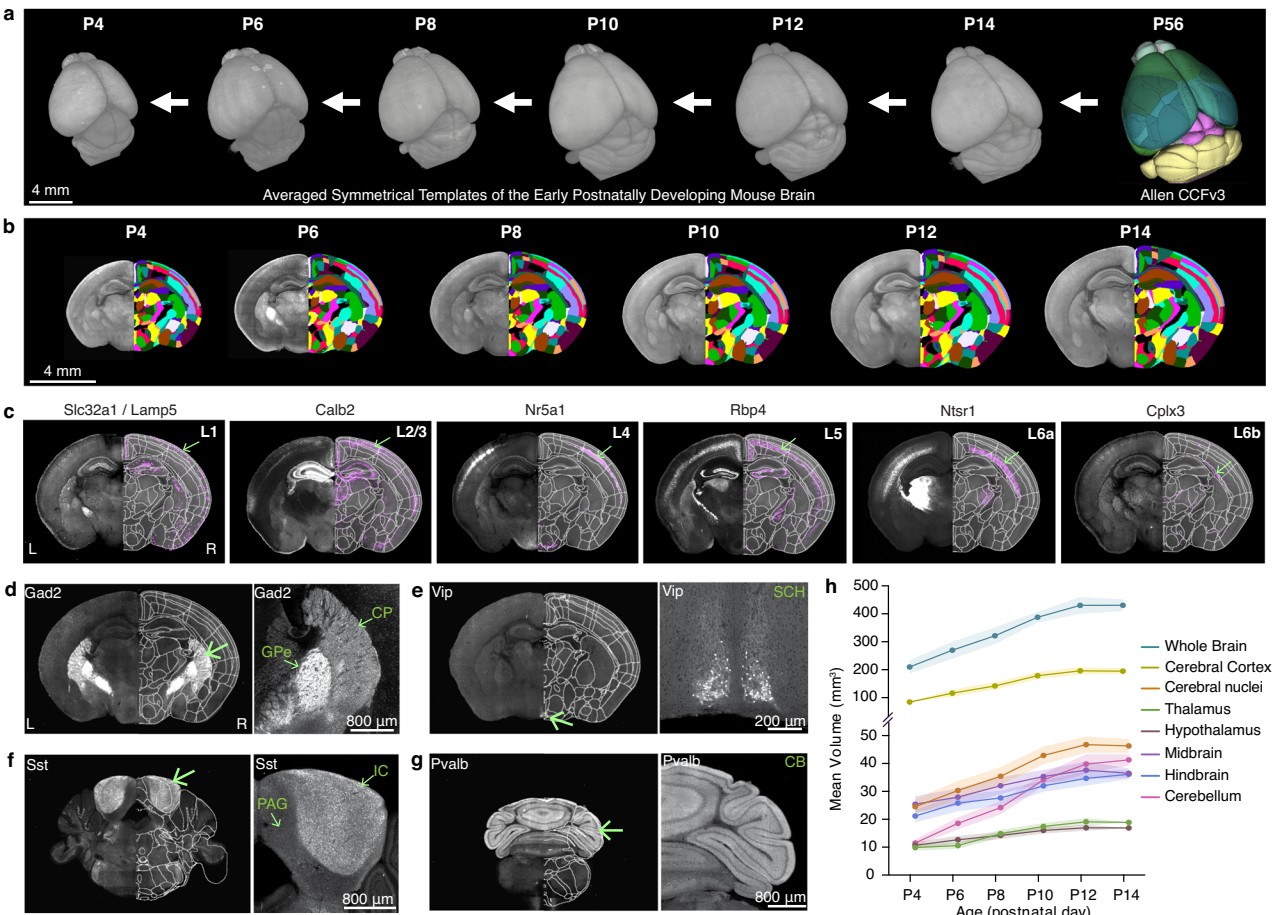

**Fig. 1 | Generation of early postnatal developmental mouse brain atlas (epDevAtlas). a** Symmetrical, morphology and intensity averaged templates of the early postnatally developing mouse brain at postnatal (P) days 4, 6, 8, 10, 12, and 14 using samples from STPT imaging. Sample sizes per template: P4 (*n* = 6), P6 (*n* = 8), P8 (*n* = 4), P10 (*n* = 7), P12 (*n* = 6), P14 (*n* = 5). **b** Anatomical labels from the adult Allen CCFv3 were registered to each developmental time point in a stepwise manner. **c** Cortical layer-specific cell subclass labeling using genetic strategies was implemented to refine and validate anatomical labels (L1 = layer 1, etc.). Cells are labeled by these transgenic lines by the abbreviated name of the driver lines

(Table 1). Representative STPT coronal brain images shown are all from P10 animals. **d–g** Examples of anatomical brain region delineations via cell type-specific labeling from (**d**) *Gad2*, (**e**) *Vip*, (**f**) *Sst*, and (**g**) *Pvalb* mice, collectively guiding epDevAtlas annotations. **h** Volumetric brain growth chart during early postnatal mouse development. All brain region volumes (mm³) are reported as the mean ± standard deviation (s.d.; shaded area between error bars) (total *n* = 38). Source data are provided as a Source Data file. Additional abbreviations: CB, cerebellum; CP, caudoputamen; GPe, external globus pallidus; IC, inferior colliculus; PAG, periaqueductal gray; SCH, suprachiasmatic nucleus.

fluorescence microscopy). Importantly, the adaptability of these epDevAtlas templates, which use the same anatomical labels as the widely utilized Allen CCFv3 for the adult mouse brain[44], enables the quantitative mapping of different cell types.

As GABAergic neurons are key in setting the inhibitory tone of the brain, we first examined how this cell class changes in space and time during the first two weeks of life in mice, which is the period when genetic and external stimuli dynamically shape brain development. Using *Gad2* mice, we applied our mapping pipeline to image the whole brain at single cell resolution and quantified the 3D distribution of labeled cells on age-matched epDevAtlas templates (Fig. 2b, c; Source Data). For each brain region depicted in the growth chart (Fig. 2c), where bubble size represents cell density, we observed temporal changes in *Gad2* cell density, which fell into three main patterns. We observed that from P4 to P14, *Gad2* cell density either 1) continually increases, 2) initially declines followed by stabilization, or 3) remains relatively stable over time. For example, in the telencephalon, *Gad2* cell densities increased markedly in the main olfactory bulb (MOB) and the striatum (e.g., caudate putamen; CP, nucleus accumbens; ACB) (Fig. 2d–g), consistent with elevated *Gad2* gene expression and continued neurogenesis in those areas during the early postnatal period[48,49]. Notably, *Gad2* in the CP is largely attributed to long-range

projecting medium spiny neurons, which transition from the striosome at P4 to the surrounding matrix at P14 following their embryonic birth timing (Fig. 2f)[50,51]. Conversely, *Gad2* cell densities were significantly reduced in the olfactory cortex (e.g., anterior olfactory nucleus; AON, piriform cortex; PIR) (Fig. 2d, e), hippocampus, and isocortex (Fig. 2h, i), where GABAergic neurons primarily function as local interneurons. Given that subpallial striatal regions receive the main excitatory inputs from the cerebral cortex, the observed elevation of *Gad2* neuronal density in the striatum and its reduction in the cortex may suggest a potential shift in inhibitory modulation within highly interconnected brain regions during early postnatal development. *Gad2* cell density in other brain areas remained relatively low and stable compared to telencephalic regions (Fig. 2c; Source Data).

**Isocortical GABAergic neurons reach adult-like patterns at P12**
Previous studies have shown that isocortical GABAergic interneurons undergo programmed cell death in an activity-dependent manner during the early postnatal period[12,14]. However, the timing of when regionally distinct GABAergic neuronal densities are established and when the population reaches stable adult-like spatiotemporal patterns remain unclear. To address this, we conducted a detailed analysis of the spatiotemporal distributions of *Gad2* cells in the isocortex.

**Table 1 | List of transgenic reporter mice**

| # | Line Name | Abbreviations | Gene | Driver Mouse Line | Reporter Mouse Line | Cell Type Labeled and Considered for Study |
|---|---|---|---|---|---|---|
| 1 | Gad2-IRES-Cre; Ai14 | Gad2 | Glutamic acid decaboxylase 2 | Gad2-IRES-Cre | Ai14 (tdTomato) | Pan-GABAergic (Gad2-expressing) neurons |
| 2 | Sst-IRES-Cre; Ai14 | Sst | Somatostatin | Sst-IRES-Cre | Ai14 (tdTomato) | Sst-expressing neurons |
| 3 | Vip-IRES-Cre; Ai14 | Vip | Vasoactive intestinal peptide | Vip-IRES-Cre | Ai14 (tdTomato) | Vip-expressing neurons |
| 4 | Pvalb-IRES-Cre; Ai14 | Pvalb | Parvalbumin | Pvalb-IRES-Cre | Ai14 (tdTomato) | Pvalb-expressing neurons |
| 5 | Slc32a1-IRES-Cre; Lamp5-P2A-FlpO; Ai65 | Slc32a1/Lamp5 | solute carrier family 32 member 1/Lysosomal Associated Membrane Protein Family Member 5 | Slc32a1-IRES-Cre;Lamp5-P2A-FlpO | Ai65 (tdTomato) | Slc32a1/Lamp5-expressing Layer 1 cortical neurons |
| 6 | Calb2-IRES-Cre; Ai14 | Calb2 | Calbindin2 | Calb2-IRES-Cre | Ai14 (tdTomato) | Calb2-expressing Layer 2/3 cortical neurons |
| 7 | Nr5a1-Cre; Ai14 | Nr5a1 | Nuclear Receptor Subfamily 5 Group A Member 1 | Nr5a1-Cre | Ai14 (tdTomato) | Nr5a1-expressing Layer 4 cortical neurons |
| 8 | Rbp4-Cre_KL100; Ai14 | Rbp4 | Retinol binding protein 4 | Rbp4-Cre_KL100 | Ai14 (tdTomato) | Rbp4-expressing Layer 5 cortical neurons |
| 9 | Ntsr1-Cre_GN220; Ai14 | Ntsr1 | Neurotensin receptor 1 | Ntsr1-Cre_GN220 | Ai14 (tdTomato) | Ntsr1-expressing Layer 6 cortical neurons |
| 10 | Cplx3-P2A-FlpO; Ai193 | Cplx3 | Complexin 3 | Cplx3-P2A-FlpO | Ai193 (Flp: tdTomato) | Cplx3-expressing Layer 6b cortical neurons |
| 11 | Cx3cr1-GFP(+/-) | Cx3cr1 | CX3C motif chemokine receptor 1 | Cx3cr1-GFP (+/-) | - | Cx3cr1-eGFP-expressing brain microglia |

In the isocortex, we observed a continuous increase in *Gad2* neuronal number from P4 until P10, likely reflecting the delayed acquisition of *Gad2* gene expression (Supplementary Fig. 2)[48], followed by a sharp decline (~20%) until P14 (Fig. 3a–c; Source Data). Throughout this period, isocortical volume showed continued growth, starting from P4, and reaching a plateau around P12 (Fig. 3c). We analyzed cell density trends using two complementary approaches: Bayesian multilevel modeling to assess density changes and functional data analysis (FDA) to examine spatial distribution patterns (See Methods). While *Gad2* density levels decreased significantly with age (Est. = −13493.42, CI [−20242.85, −6588.72]), FDA results indicated that the spatial distribution pattern remained stable over time ($F = 2.08$, permuted p = 0.0957). We found that *Gad2* cell density was highest at P4, experienced a significant decline at P10, and reached a stable and adult level at P12 (Fig. 3d, e; Supplementary Fig. 3a). These data are concordant with the established notion that developmentally regulated apoptosis of GABAergic cortical interneurons takes place between P1 and P15, with peak programmed cell death occurring between P7 and P11[12]. Previously we observed that GABAergic neurons are more densely expressed in sensory cortices compared to association areas in adult mice[31,52]. To examine the emergence of regionally distinct *Gad2* cell densities, we utilized our isocortical flatmap, which provides anatomical delineations, along with five distinct cortical domains based on connectivity, each represented by a different color (Fig. 3f)[52,53]. We found that *Gad2* cell density was highly enriched in sensory cortical regions (e.g., somatosensory; SS, auditory; AUD, and visual; VIS) and relatively low in association cortices (e.g., prelimbic area; PL), a pattern evident as early as P4 and sustained to P14 (Fig. 3d–g). Cortical layer (L)-specific analysis revealed that *Gad2* neurons in L2/3 and L5 exhibited higher densities compared to other layers at P4, which was maintained until P14 compared to other layers (Supplementary Fig. 3d–k). Lastly, we confirmed the specificity of our *Gad2*-Cre;Ai14 mouse line at ages P4 and P10 using fluorescence in situ hybridization (FISH) for *Gad2* and tdTomato and found that both gene expression and reporter expression closely correspond with each other (Fig. 3h, i).

### *Sst* and *Vip* cortical interneuron subclasses undergo differential growth patterns

*Sst* and *Vip* are key neuropeptides in maturing cortical interneuron subclasses that increase their expression during early postnatal period to support circuit maturation[54–58]. Disruption of *Sst* and *Vip* expression has been implicated in neurodevelopmental disorders[59,60]. Here, we examine how *Sst* and *Vip* interneuron subclasses populate different cortical areas during the early postnatal period using *Sst*-Cre or *Vip*-Cre mice crossed with Ai14 reporter mice, respectively[11,61].

We observed that the number of *Sst* interneurons steadily increased from P4 to P10 and then leveled off until P14 (Fig. 4a–c; Source Data). Overall density of *Sst* interneurons gradually decreased over time from P4 to P14 (global test: Est. = −4395.22, CI [−6170.32, −2612.19]), with earlier time points (P4 and P6) showing higher densities than later ages (P10-P14 and P56) (Fig. 4d–f; Supplementary Fig. 3b). Notably, we found regionally distinct *Sst* density reductions ($F = 3.61$, permuted $p = 0.0103$) (Fig. 4d–f), in contrast to the relatively even reduction observed in *Gad2* density across different cortical areas (Fig. 3d). For instance, *Sst* neuronal densities in medial (e.g., infralimbic: ILA) and lateral association cortices (e.g., temporal association area; TEa, perirhinal area; PERI, agranular insular area; AI) showed a dramatic reduction, where they were most enriched at P4 (Fig. 4b, d, e). On the other hand, the somatosensory cortex (SS) displayed the least change (Fig. 4d, e). When considering cortical layers, *Sst* neuronal densities in L5 and L6 were the highest as early as P4, and they sharply decreased in lateral association cortices, while minimal changes were observed in superficial layers (Supplementary Fig. 3l–q).

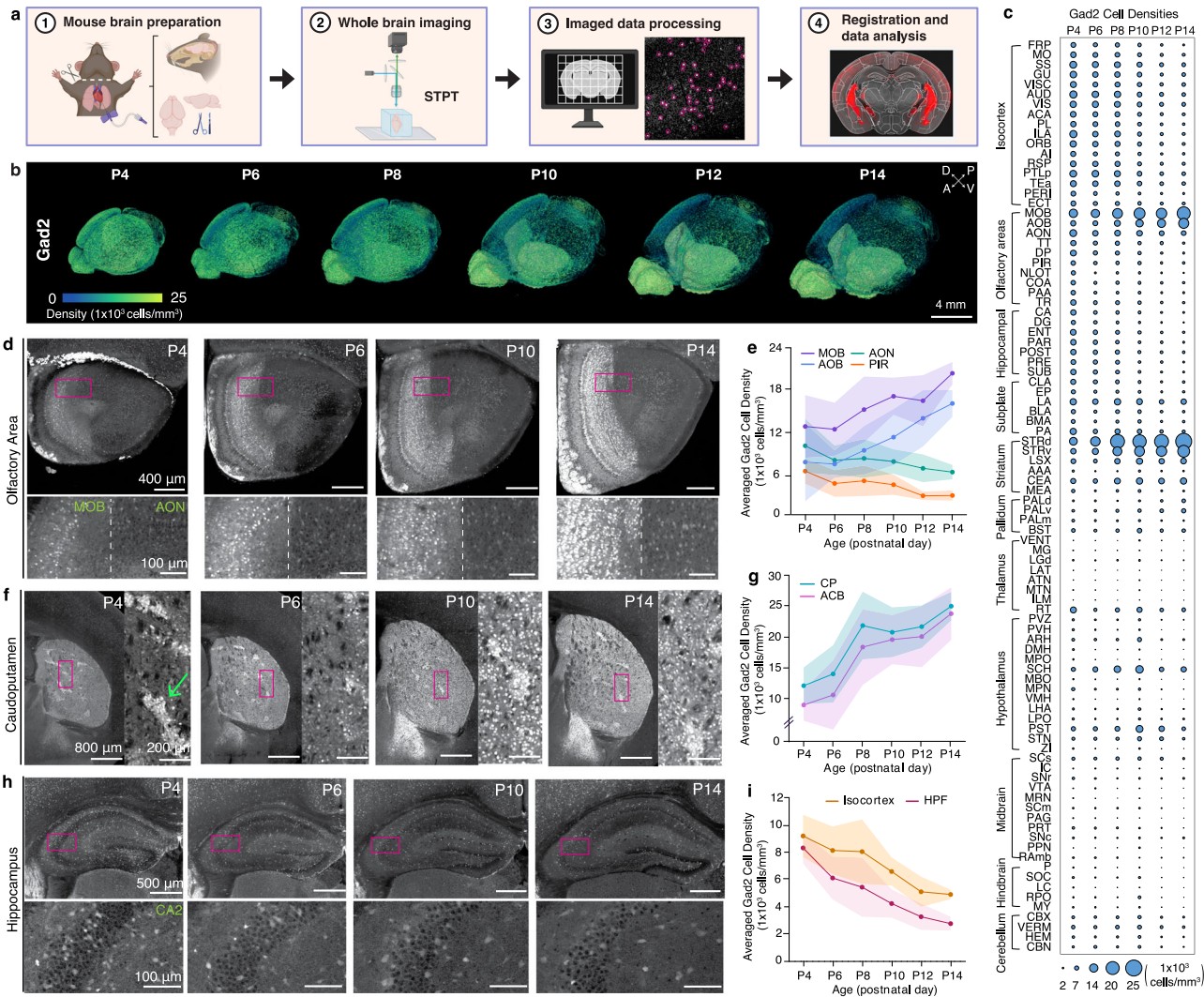

**Fig. 2 | Brain-wide mapping of early postnatally developing GABAergic neurons. a** Overview of cell type mapping pipeline. **b** 3D renderings of *Gad2* cell density (cells/mm³) registered to age-matched epDevAtlas. Each processed image per age is from a representative *Gad2*-IRES-Cre;Ai14 brain sample. **c** *Gad2* cell type growth chart across early postnatal mouse brain development, emphasizing major brain regions based on the anatomical hierarchy of Allen CCFv3. **d** Representative STPT images of *Gad2* cells and **e** their average density in the olfactory brain regions including main olfactory bulb (MOB), accessory olfactory bulb (AOB), anterior olfactory nucleus (AON), and piriform cortex (PIR) between P4 and 14. **f,g** Striatal

GABAergic cells display increased *Gad2* expression from P4 to P14. **f** Representative STPT images of *Gad2* cells and **g** their density in the caudoputamen (CP), with striosomes (green arrow) exhibiting earlier maturation, and nucleus accumbens (ACB). **h** Representative STPT images and **i** density of *Gad2* cells in the hippocampal formation (HPF) and isocortex. All data are reported as mean ± s.d. (includes shaded areas between error bars). Additionally, regions selected in Fig. 2d–i represent those with the most notable density trends, while a complete dataset for all regions is available in Fig. 2c. Source data are provided as a Source Data file. Figure 2a was created in BioRender. Kim, Y. (2025) https://BioRender.com/lriwwxt.

In contrast to *Sst* interneurons, *Vip* interneuron cell density was constant between P4 and P6, followed by an increase from P6 to P8, and then reached stability from P8 and onward until P56 (Fig. 4l–m; Supplementary Fig. 3c, Source Data). The number of cortical *Vip* cells rapidly increased from P4 to P12 with cortical volume increase, resulting in significant cell density increase between P6 and P8 (global test: Est. = 919.12, CI [304.25, 1537.81]) (Fig. 4i–n), likely reflecting the delayed acquisition of *Vip* gene expression (Supplementary Fig. 2). *Vip* neurons are primarily enriched in the medial association (e.g., retrosplenial cortex; RSP) and audio-visual domains of the isocortical flatmap while overall density pattern remains stable across the postnatal period (Fig. 4l–m). *Vip* cell density was highest in L2/3 throughout the early postnatal weeks, with a relatively stable trajectory across all regions except for the lateral association cortices, in which L2/3 *Vip* expression was lowest at P4 and P6 before increasing until P12 (Supplementary Fig. 3r–w).

FISH validation of *Sst*-Cre;Ai14 and *Vip*-Cre;Ai14 mouse lines at ages P4 and P10 showed that tdTomato expression overlaps with target cell type markers at about 80–85% (Fig. 4g, h, o, and p). These results confirm that gene expression closely corresponds to reporter expression.

Overall, these findings suggest that GABAergic interneuron subclasses may display spatiotemporal heterogeneity based on their developmental origins and birthdates.

## Microglial expansion exhibits regional heterogeneity in the early postnatal mouse brain

Microglia play a pivotal role in mediating the programmed cell death of neurons and facilitating their maturation during early postnatal development[62,63]. However, it remains unclear how microglial density evolves during early postnatal development and its connection to the early postnatal trajectory of GABAergic cells across various brain

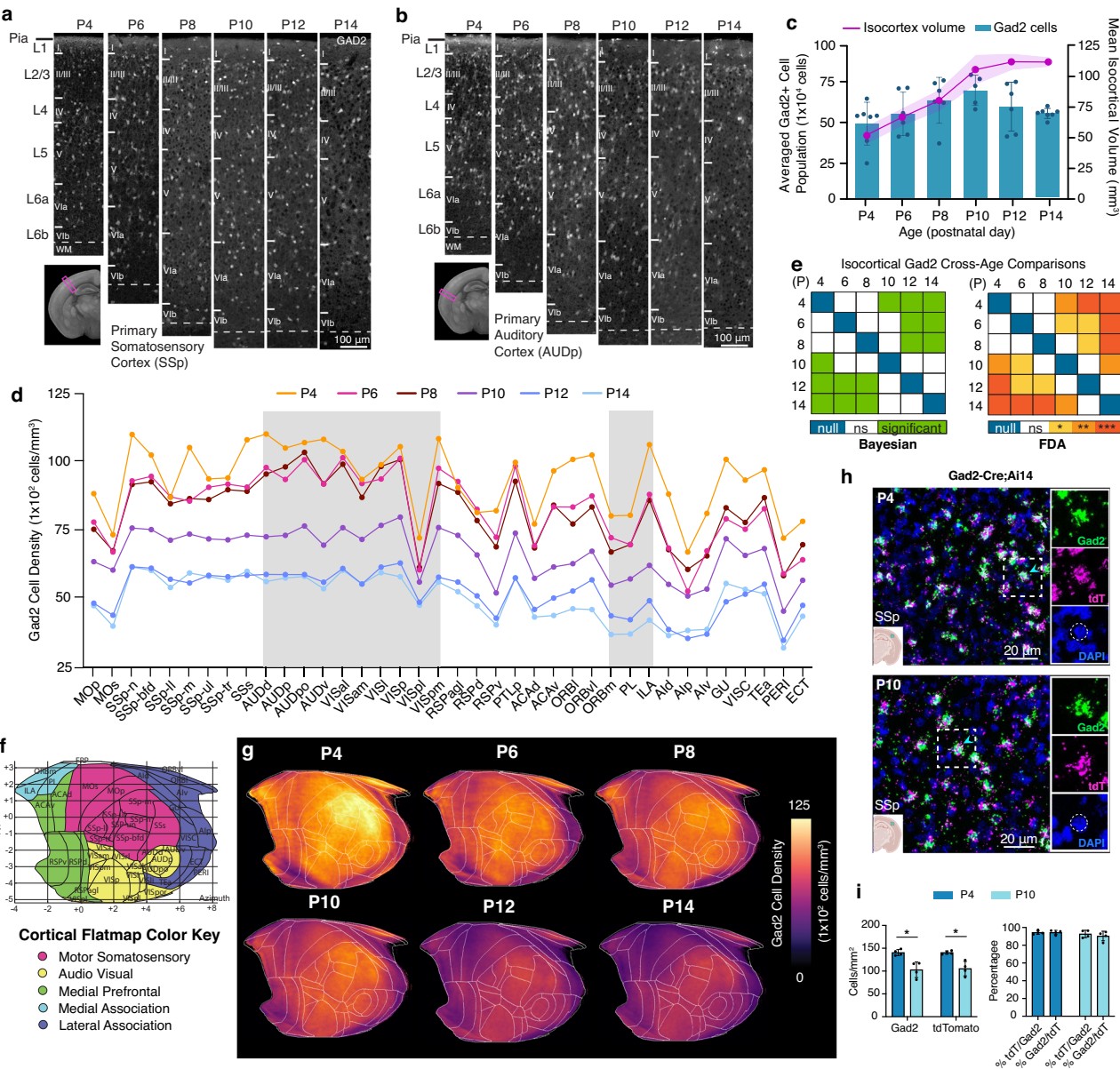

**Fig. 3 | GABAergic neuronal development in the isocortex. a,b** Representative STPT images of *Gad2* cells in the **a** primary somatosensory cortex (SSp) and **b** primary auditory cortex (AUDp). **c** Temporal trajectory of *Gad2* cell count vs. volume (mm³) in the isocortex. All data are reported as mean ± s.d. (includes shaded area between error bars). **d** Averaged *Gad2* cell density patterns across isocortical areas. Each isocortical brain region falls into one of five categories, grouped by their previously known functional and anatomical connectivity profiles. **e** (left) Bayesian multilevel modeling, with a global hypothesis test and post hoc pairwise comparisons, was used to examine significant magnitude differences in density of Gad2 cells between all age pairs across isocortical regions (null, 0; ns, non-significant; significant). See Supplementary Data 1 for Estimates and Highest Posterior Density (HPD) intervals. (right) Functional data analysis (FDA) to examine significant differences between density patterns of isocortical *Gad2* cells between all age pairs (null, 0; ns, non-significant; two-tailed Student's t-test: *$p < 0.05$;

**$p < 0.005$, ***$p < 0.001$, ****$p < 0.0001$). See Supplementary Data 1 for exact p-values. **f** Isocortical flatmap with Allen CCFv3 anatomical regions and border lines. The y-axis represents the bregma's anterior-posterior (AP) coordinates, while the x-axis indicates azimuth coordinates to combine medial-lateral and dorsal-ventral direction. **g** Isocortical flatmaps of 3D counted and averaged *Gad2* cell densities. **h** Confocal image of a cropped primary somatosensory region after fluorescence in situ hybridization, demonstrating colocalization of *Gad2* (green), tdTomato (magenta), and DAPI (blue). **i** Quantification of individual and colocalized *Gad2*-positive and tdTomato-positive cells at P4 and P10 (total n = 8 mice, *n* = 4 mouse brain sections per age). P-values (*p* = 0.0159 for *Gad2*-positive cells between P4 and P10; *p* = 0.0180 for tdTomato-positive cells between P4 and P10) were obtained using a two-tailed Student's t-test. All data are reported as mean ± s.d. (including the error bars). See Source Data for *Gad2* cell counts, density, volume measurements, and abbreviations, all of which are provided as a Source Data file.

regions. To address this, we employed heterozygous *Cx3cr1*-eGFP$^{(+/-)}$ reporter mice and harnessed the epDevAtlas to systematically quantify and examine microglial distributions in the postnatally developing mouse brain.

Our comprehensive cell density mapping results unveiled significant spatial variations in the distribution of *Cx3cr1* microglial

populations during their early postnatal colonization of the CNS, from P4 to P14 (Fig. 5a, b; Source Data). Of note, we observed the accumulation of proliferating microglia in the corpus callosum (Fig. 5a, c) and the cerebellar white matter (Fig. 5a, d). The clusters of white matter-localized microglia were evident until P8 and experienced a sudden population decline by P10 (Fig. 5a, c, d, g). In the cerebellum, microglia

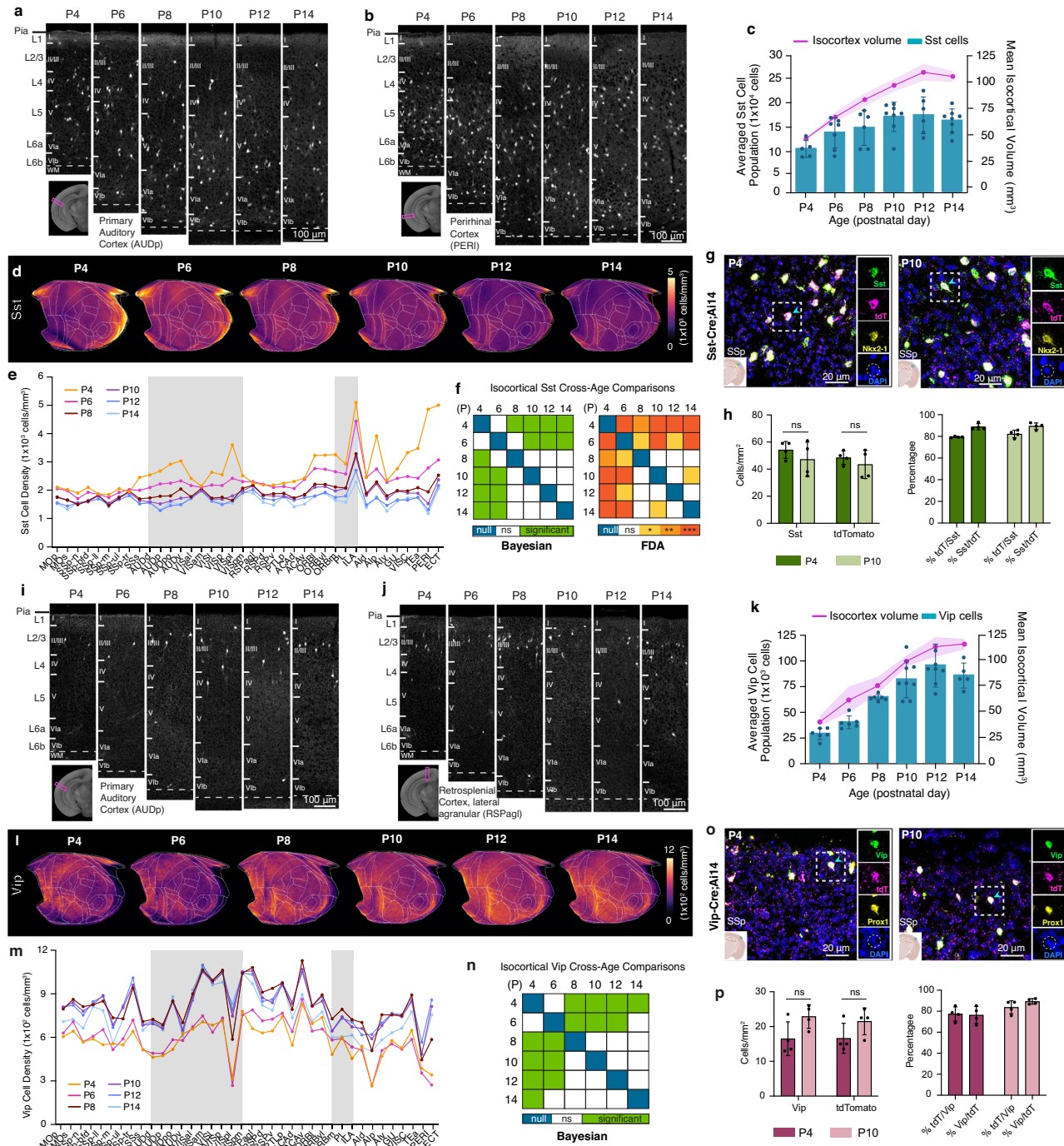

**Fig. 4 | Cortical GABAergic cell types undergo differential developmental trajectories. a,b** Representative STPT images of *Sst* cells in the **a** primary auditory cortex (AUDp) and **b** perirhinal cortex (PERI). **c** Temporal trajectory of *Sst* cell count vs. volume in the isocortex. Data are reported as mean ± s.d. (includes shaded area between error bars). **d** Isocortical flatmaps of *Sst* cell densities. **e** Averaged *Sst* cell density across isocortical areas. **f** (left) Significant differences in *Sst* densities across age groups were identified using Bayesian global tests and pairwise comparisons. See Supplementary Data 1 for estimates and HPD intervals. (right) Functional data analysis (FDA) to examine significant differences between density patterns of isocortical *Sst* cells between all age pairs (null, 0; ns, non-significant; two-tailed Student's t-test: *p < 0.05; **p < 0.005; ***p < 0.001; ****p < 0.0001). **g** Confocal image of a cropped SSp region after fluorescence in situ hybridization, demonstrating colocalization of *Sst* (green), tdTomato (magenta), *Nkx2-1* (yellow), and DAPI (blue). **h** Quantification of individual and colocalized *Sst*-positive and tdTomato-positive cells at P4 and P10 (total *n* = 8 mice, *n* = 4 mouse brain sections per age). P-values were obtained using a two-tailed Student's t-test. All data are reported as mean ±

s.d. (including the error bars). **i,j** Representative STPT images of *Vip* cells in the **i** primary auditory cortex (AUDp) and **j** retrosplenial cortex, agranular area (RSPagl). **k** Temporal trajectory of *Vip* cell count vs. volume. Data are reported as mean ± s.d. (includes shaded area between error bars). **l** Isocortical flatmaps of *Vip* cell densities. **m** Averaged *Vip* cell density (cells/mm³) patterns across isocortical areas. **n** Bayesian global test and pairwise comparisons identified significant density differences across *Vip* age groups (See Supplementary Data 1 for estimates and HPD intervals). **o** Confocal image of a cropped SSp region after RNAscope, demonstrating colocalization of *Vip* (green), tdTomato (magenta), *Prox1* (yellow), and DAPI (blue). **p** Quantification of individual and colocalized *Vip*-positive and tdTomato-positive cells at P4 and P10 (total *n* = 8 mice, *n* = 4 mouse brain sections per age). P-values were obtained using a two-tailed Student's t-test. All data are reported as mean ± s.d. (including the error bars). See Source Data for cell counts, density, volume measurements, and abbreviations, all of which are provided as a Source Data file for both *Sst* and *Vip* cells.

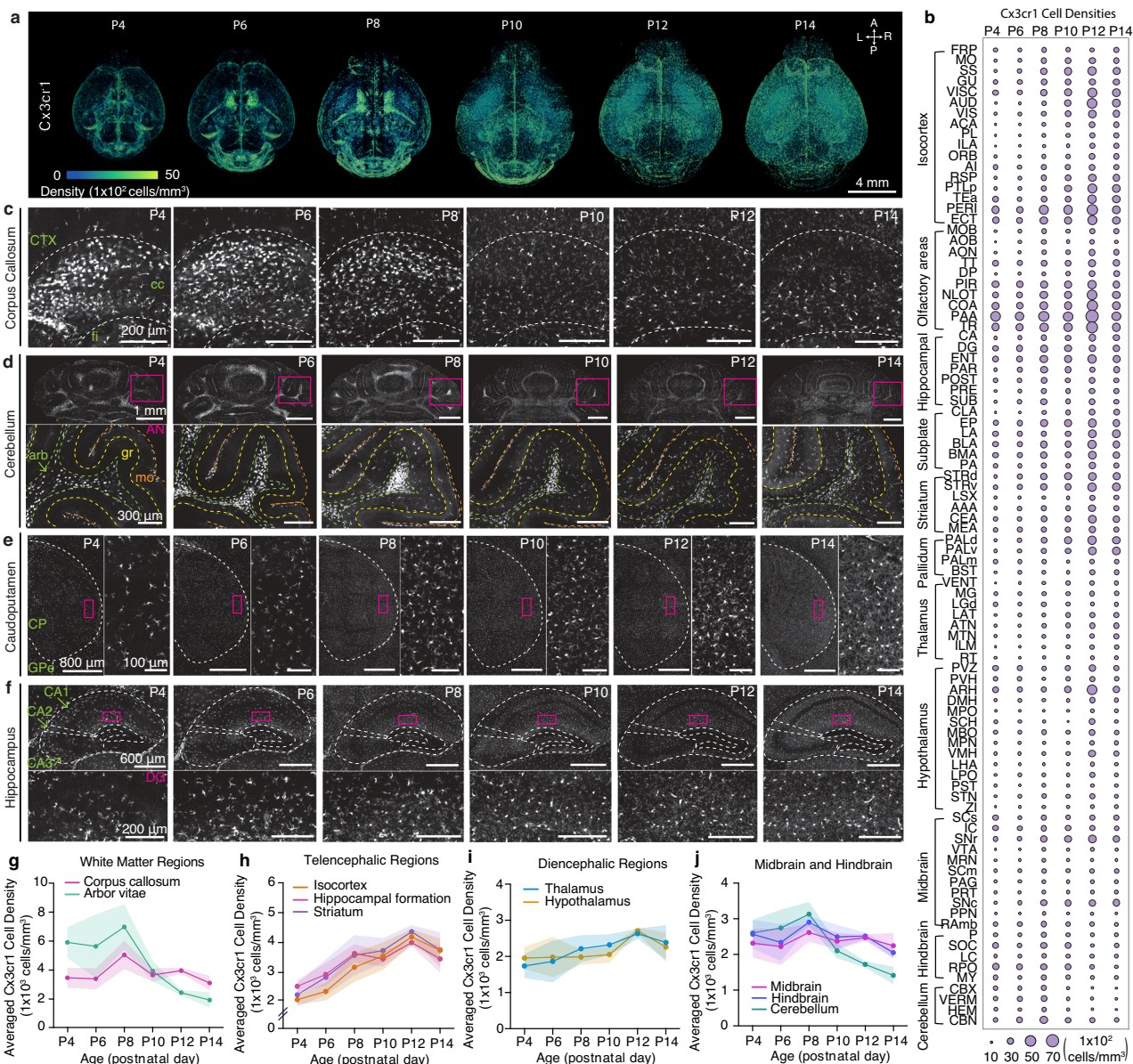

**Fig. 5 | Brain-wide mapping of early postnatally developing microglia. a** 3D renderings of *Cx3cr1* microglial cell density from representative *Cx3cr1*-eGFP[(+/−)] samples registered to age-matched epDevAtlas. **b** Microglial (*Cx3cr1*) cell type growth chart **c–f** Representative STPT images of *Cx3cr1* microglia in the corpus callosum (cc), **d** the cerebellum, **e** the caudopatamen, and **f** the hippocampus. Inset images in (**d-e**) are high magnification images from magenta-colored boxed areas.

**g–j** Averaged *Cx3cr1* microglial density in (**g**) white matter, specifically the corpus callosum and arbor vitae of the cerebellum, **h** telencephalic regions, **i** diencephalic regions, and **j** midbrain and hindbrain regions. Data are reported as mean ± s.d. (shaded area between error bars). See Source Data for *Cx3cr1* microglial cell counts, density, volume measurements, and abbreviations, all of which are provided as a Source Data file.

are also enriched in white matter until around P10, before progressively spreading to cover other layers of the cerebellar cortex (Fig. 5d). Additionally, microglial morphology dynamically changes during the early postnatal period. Up until P8, microglia exhibited ameboid morphology with thick primary branches and larger cell bodies, distinct from the more ramified microglia with comparably smaller somas observed from P10 onwards (Supplementary Fig. 4a–d). This morphological alteration between the first and second postnatal weeks is even more substantial in the white matter and cerebellum, implying that as white matter tracts and the cerebellar cortex mature, microglia gradually transition into a more complex and ramified phenotype (Fig. 5c, d, Supplementary Fig. 4a, b). By P12 and P14, these clonal microglia displayed a more dispersed distribution across the entire brain, forming a mosaic pattern reminiscent of tessellation (Fig. 5a,

c–f). This mosaic-like distribution achieved by the end of the second postnatal week is then maintained into adulthood, although regional heterogeneity exists[37,64].

In telencephalic regions, encompassing both cortical and striatal areas, *Cx3cr1* microglia displayed the most rapid expansion in density, with an approximately 200% increase from P4 to P14 (Fig. 5e, f, h). For example, in the striatum and hippocampus, we observed the largest increases in microglial density at the transition between the first and second postnatal weeks, with continued gradual increase until the end of the second week (Fig. 5e, f, h). Diencephalic regions, including the hypothalamus and thalamus, displayed modest increases in microglial density, ranging from about 50% to 100% between P4 and P14 (Fig. 5i). In contrast, microglial density in the midbrain and hindbrain remained relatively stable from P4 to P14, while the cerebellum exhibited a sharp

decrease at the end of the second postnatal week with the significant reduction of white matter-localized microglia in the arbor vitae (arb) region (Fig. 5d, g, j; Source Data). It is also worth noting that the change in microglial morphology from ameboid to ramified in the corpus callosum and cerebellar-related fiber tracts aligns with the drop in microglial cell density at P8 (Fig. 5g, Supplementary Fig. 4a, b).

In summary, our results demonstrate that microglia undergo an initial expansion in selected white matter tracks, followed by further colonization of the gray matter at different rates across distinct brain regions.

### Sensory processing cortices and the dorsal striatum exhibit high microglial density

We next examined the detailed spatiotemporal patterns of microglia in the isocortex, considering their vital roles in fine-tuning maturing cortical inhibitory circuits[20,21]. We observed a rapid increase in the number of cortical *Cx3cr1* microglia, outpacing the growth in isocortex volume (Fig. 6a–c). The average density of isocortical *Cx3cr1* microglia rose swiftly between P4 and P12 (global test: Est. = 6429.61, CI [4476.02, 8384.94]), showing a strong negative correlation with isocortical *Gad2* interneuron density (Pearson's correlation coefficient: $r = -0.9189$, $p < 0.0096$) (Fig. 6d–f). After P12, however, microglial density showed a gradual decline until P56, due to developmentally related microglial apoptosis and a decreased proliferative capacity (Supplementary Fig. 4g)[65]. Our investigation also revealed a regionally heterogeneous expansion of microglia ($F = 8.68$, permuted $p = 0.0001$) (Fig. 6e, f). To visualize these spatial density patterns of microglia over time, we employed the isocortical flatmap (Fig. 6g). At P4, the flatmap highlighted the emergence of heightened *Cx3cr1* density in specific focal areas within the medial and lateral association regions linked with white matter-localized microglia. By P6, there was increased microglial density within sensory regions, particularly in the primary somatosensory cortex (SSp) and the retrosplenial cortex (RSP) (Fig. 6e, g). At P12 and P14, microglia began to densely populate other sensory areas, including the auditory (AUD) and visual (VIS) cortices (Fig. 6e, g). This creates a gradient of low- to high-density microglial distribution along the anterior-posterior axis in the isocortex by the conclusion of the second postnatal week (Fig. 6g). Overall, these findings from the isocortex strongly suggest that the emergence of high-density microglia in sensory cortices is closely associated with the onset of active sensory input, such as active whisking (linked with SSp) and the vestibular righting reflex (linked with RSP) around P6, as well as the opening of ears (AUD) and eyes (VIS) around P12[66–68].

Considering that the caudate putamen (CP), also known as the dorsal striatum, receives topological projections from distinct cortical areas[69], we questioned whether microglial density changes in the CP resemble developmental patterns observed in the isocortex. Indeed, we found that the ventrolateral CP (CPvl), which primarily receives projections from the SSp, exhibited a higher density compared to the ventromedial CP (CPvm), which receives projections from the association cortex (Fig. 6h, i)[69]. This difference in CP subregional density was not evident at P4 and began to significantly increase at P8, aligning with our results showing that microglial density starts to prominently populate the SSp at this time (Fig. 6g, h; Source Data). During the second postnatal week, we observed a further increase in CPvl microglial density compared to other CP subregions, which was significant at P12 (Fig. 6h; Source Data).

These findings suggest that microglia preferentially increase their density in sensory processing areas, despite their broad proliferation and allometric expansion patterns throughout the developing brain.

### Cell type growth chart as a new resource

To enhance accessibility to epDevAtlas and its detailed cell type mappings, we developed a user-friendly web visualization based on Neuroglancer, available at https://kimlab.io/brain-map/epDevAtlas.

This platform allows users to explore full-resolution images and mapped cell density data with age-matched epDevAtlas templates and labels, including cortical layer-specific reporter mice (Fig. 7a–d).

For instance, we used *Nr5a1*-Cre;Ai14 mice to label L4 cortical neurons. We observed an early emergence of labeled cells in the SS region of the developing cortex at P6, followed by a remarkable surge in cell density within L4 of the barrel field (SSp-bfd) at P10 that continually increased until P14 (Fig. 7e, g; Source Data). In comparison, *Nr5a1* cells within L4 of AUD and VIS regions exhibited a delay in density growth with sudden increase at P14 (Fig. 7f, g; Source Data). This observation strongly indicates a correlation between *Nr5a1* expression in cortical L4 and the developmental onset of individual sensory modalities.

Moreover, the integration of multiple cell types allows us to pinpoint regional variations in cell type compositions. Take P4 as an example, where we observed that GABAergic neurons and microglia displayed contrasting yet complementary density patterns in the isocortex and olfactory cortex (Fig. 7h). While GABAergic neurons maintained higher densities in the isocortex compared to the olfactory cortex, microglia exhibited the opposite pattern (Fig. 7h).

With this new resource, users can explore individual cell types or their combinations, facilitating comparisons of their spatial distribution across developmental stages, as summarized in the example given for the isocortex (Fig. 7i; Source Data).

## Discussion

We present cell type growth charts of GABAergic neurons and microglia in the early postnatally developing mouse brain using the epDevAtlas as 3D STPT-based atlases. Standard biological growth charts are essential tools to comprehend normal growth and identify potential pathological deviations[1,70]. Existing brain growth charts are largely limited to macroscopic volumetric and shape analyses. Therefore, our novel cell type growth charts significantly enhance our understanding of brain cell type composition during early development and can serve as the standard metric for evaluating alterations in pathological conditions.

The importance of 3D brain atlases as a standardized spatial framework is well recognized in integrating diverse cell census information[71,72]. For instance, the Allen CCFv3 serves as a widely used standard adult mouse brain atlas for cell census data integration[44,73]. However, the lack of a similar atlas for developing brains has hindered the systematic examination of different cell types and their evolution across neurodevelopment. Although emerging atlas frameworks have become available for the developmental mouse brain[39,40,74], anatomical labels with different ontologies and sparse developmental time points create significant challenges in consistently interpreting cell type specific signals, especially since the structure of the developing mouse brain rapidly evolves. Our epDevAtlas resolves this by offering morphologically and intensity-averaged symmetric templates at six key ages during the critical early postnatal period, ranging from P4 to P14. Furthermore, the epDevAtlas includes 3D anatomical annotations derived from the Allen CCFv3, validated and refined using cell type-specific transgenic animals. Hence, this fills the critical need to systematically study cell type changes in early postnatal development. Leveraging image registration tools[75–77], there is considerable potential for data from different 3D and 2D imaging modalities to be mapped onto the epDevAtlas to unveil molecular and cellular changes in developing brains. Registration of images acquired by light sheet fluorescence microscopy (LSFM) with tissue clearing, for example, was not tested using the STPT epDevAtlas templates in the current study. However, this is an area we are actively working on and plan to explore further in future research.

Applying our new atlases and mapping pipelines, we present detailed growth charts for GABAergic neurons and microglia, accounting for the regionally distinct volumetric expansion of the

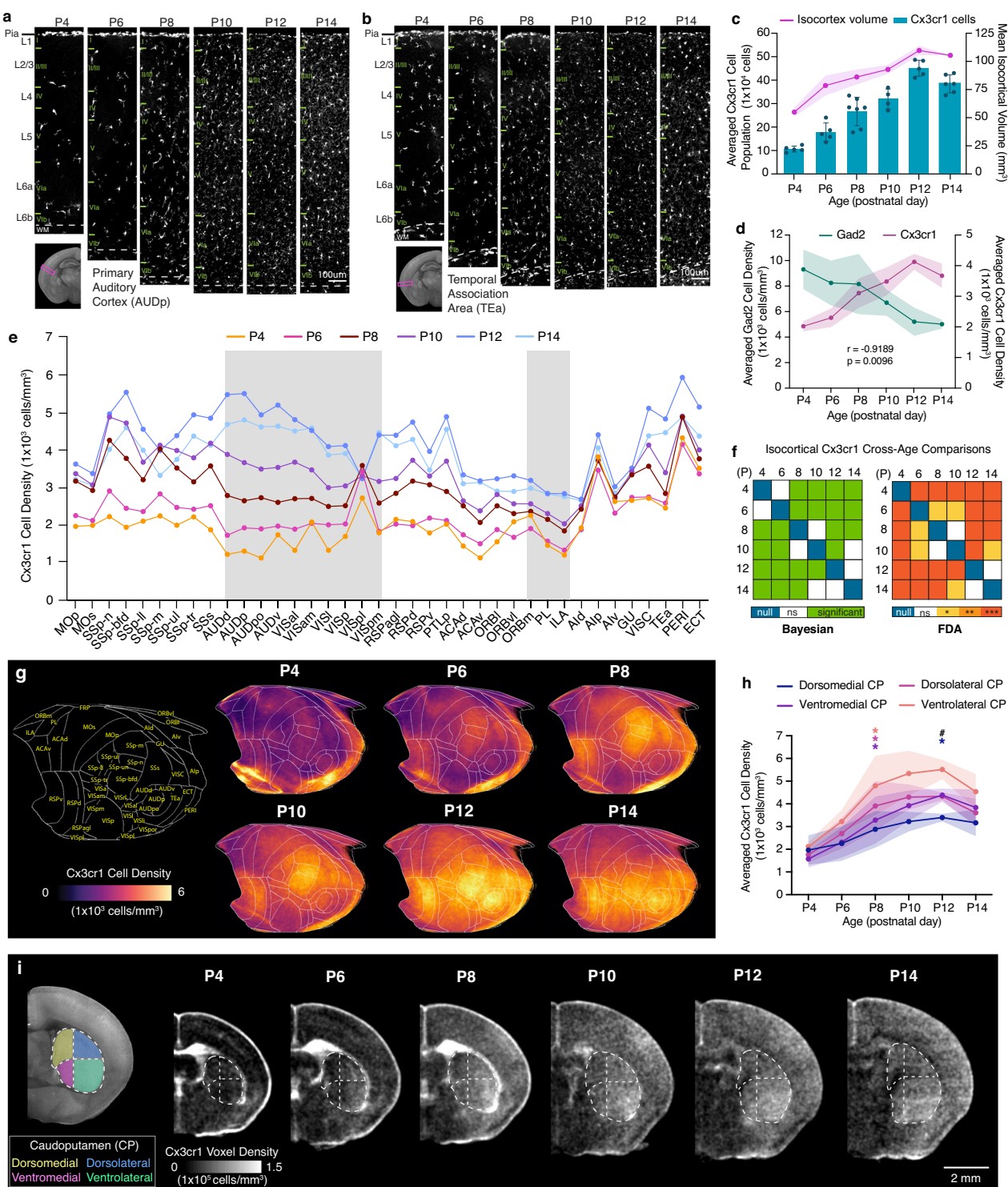

brain. We found that rapid changes in volume and cell type density, including GABAergic neuron subclasses and microglia, stabilize around P12 during the first two postnatal weeks. Earlier research has demonstrated that cortical GABAergic interneurons undergo activity-dependent cell death during early postnatal periods to reach stable densities for mature inhibitory circuits[12–14,16] and have up to two-fold differences in their density across different areas in adult brains[31,52]. We identify that this overall regional density difference is established as early as P4 and the density decreases approximately two-fold, reaching stability at P12. Additionally, we found that GABAergic interneuron classes from different developmental origins

can exhibit varying magnitudes of cell density changes[12,14]. MGE-derived *Sst* neurons showed more than a two-fold density reduction in select lateral association areas, while cell density reduction occurred at a much smaller magnitude in the SSp, suggesting regional heterogeneity in programmed cell death. On the contrary, CGE-derived *Vip* neurons established stable distribution patterns at P8, with minimal density changes until P14. A previous study showed that *Vip* neurons do not undergo activity-dependent apoptosis[14]. This evidence suggests that *Sst* neurons are more adaptive, in that they establish their mature density based on external stimuli, compared to *Vip* neurons[12,14,78,79].

**Fig. 6 | Cortical microglial expansion during early postnatal development. a-b** Representative STPT images of *Cx3cr1* microglia in the **a** primary auditory cortex (AUDp) and **b** temporal association cortex (TEa). **c** Temporal trajectory of *Cx3cr1* microglial count vs. volume. Data are reported as mean ± s.d. (includes shaded area between error bars). **d** Microglial (*Cx3cr1*) and GABAergic (*Gad2*) cell density trajectories between P4 and P14 are significantly anti-correlated, as indicated by the calculated Pearson correlation coefficient (r = −0.9189, p < 0.0096 (two-tailed Student's t-test)). Data are reported as mean ± s.d. (includes shaded area between error bars). **e** Averaged *Cx3cr1* microglial density patterns across isocortical areas. **f** (left) Age-related variations in microglia densities were assessed using Bayesian global tests and pairwise comparisons. Detailed estimates and HPD intervals are available in Supplementary Data 1. (right) Functional data analysis (FDA) to examine significant differences between density patterns of isocortical *Cx3cr1* cells between all age pairs (null, 0; ns, non-significant; two-tailed Student's t-test: *p < 0.05; **p < 0.005, ***p < 0.001, ****p < 0.0001). **g** Isocortical flatmaps of *Cx3cr1* microglial

densities. **h** Microglial density patterns across four major striatal divisions of the caudoputamen (CP) from P4 to P14: dorsomedial CP (CPdm), dorsolateral CP (CPdl), ventromedial CP (CPvm), ventrolateral CP (CPvl). Color-coded asterisks (*) corresponding to CP subdivision indicate the time of emergence of significant microglial density differences. *p = 0.0191 for ventrolateral CP at P8, *p = 0.0155 for dorsolateral CP at P8, *p = 0.0447 for ventromedial CP at P8, and *p = 0.0434 for dorsomedial CP at P12. These p-values are adjusted after performing Dunnett's T3 multiple comparisons test. Hash sign (#) indicates significance in microglial density differences between CP subdivisions. Data are reported as mean ± s.d. (includes shaded area between error bars). **i** Anatomical layout of the four functional domains (dorsomedial, yellow; dorsolateral, blue; ventromedial, magenta; ventrolateral, green) show a distinct *Cx3cr1* microglial population increase in the ventrolateral CP from P4 to P14. See Source Data for *Cx3cr1* microglial cell counts, density, volume measurements, and abbreviations.

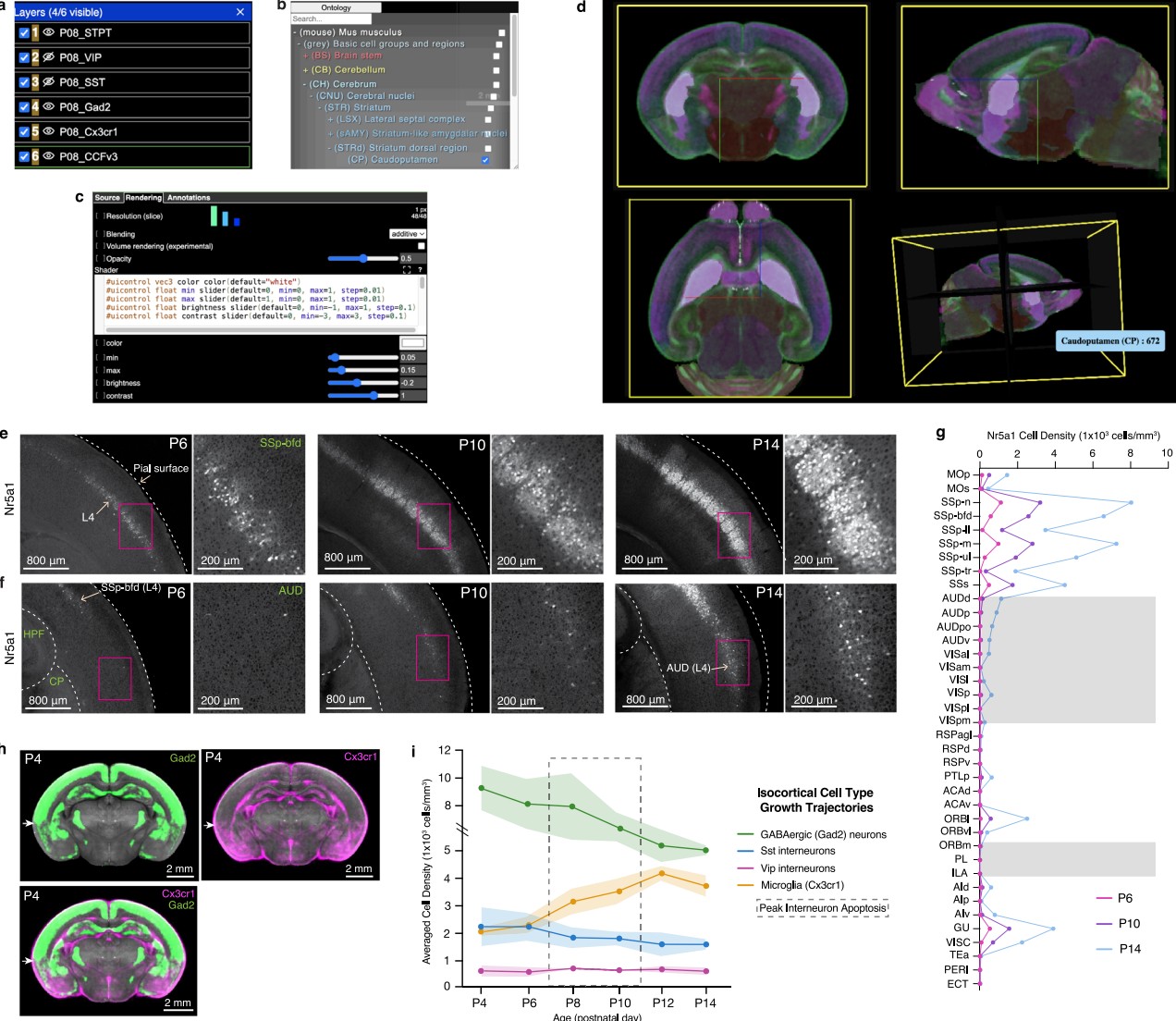

**Fig. 7 | Web visualization and integrative analysis with cell type growth charts. a–d** Neuroglancer based visualization enables users to **a** select cell types or atlas/ annotations, **b** examine anatomical label ontology, **c** modify visualization setting (e.g., color, intensity), **d** with orthogonal 3D views. **e-f** Representative STPT images of *Nr5a1* cells in layer 4 (L4) of the **e** primary somatosensory cortex, barrel fields, and **f** auditory cortex. Full resolution STPT data is accessible and viewable via online resource. **g,** *Nr5a1* cell density across isocortical areas. **h**, Average 3D density data of

*Gad2* cells (top left; green; *n* = 7) and *Cx3cr1* microglia (top right; in magenta; *n* = 5), overlaid (bottom left) registered to the P4 epDevAtlas template (gray background). White arrows highlight the border between isocortical and the olfactory cortex from the two cell types with the complementary density pattern. **i** Summary growth chart of GABAergic and microglial cell densities in the isocortex. Data are presented as mean values +/- s.d. (includes shaded area between error bars). See Source Data for *Nr5a1* cell counts, density, volume measurements, and abbreviations.

Beyond the isocortex, our results provide a comprehensive resource for examining quantitative cell type changes in other brain regions across time. Unlike cortical areas, including the hippocampus and olfactory cortices, the olfactory bulb (OB) and the striatum demonstrate continuously increasing density of *Gad2* neurons, partially due to ongoing neurogenesis during the early postnatal period[11,80,81]. Moreover, the majority of neurogenesis in the striatum is complete by birth, indicating that the rapid increase in striatal *Gad2* neuronal density represents a delayed onset of Gad2 expression in medium spiny neurons, a major neuronal subclass with long-range projections[51]. Notably, we also observed the emergence of *Gad2* expression in striosomes, followed by its increase in surrounding matrix compartments. This finding aligns with the early development of striosomes during embryonic development[50,51]. In contrast, we found an approximately two-fold decrease of *Sst* neuronal density, as one of the main interneuron types in the dorsal striatum (Source Data)[51], suggesting that the interneuron population in both cortical and striatal regions undergo significant density reductions during the early postnatal period.

As for microglia, their density in the isocortex increases about two-fold to facilitate the clearance of apoptotic neurons and promote circuit maturation[82]. Isocortical microglial density rapidly increases and peaks at P12, with the most substantial density change occurring in sensory cortices, while association areas showed a relatively smaller increase. This suggests that microglia may play an active role in shaping activity-dependent cortical development and neural circuit maturation based on external stimuli during the first two postnatal weeks[62,82,83]. The coordinated and selective increase in microglial density within the ventrolateral area of the dorsal striatum, which receives major somatosensory cortical projections, further corroborates the notion that microglia may increase its density in sensory processing areas to facilitate activity-dependent brain development. Previous studies suggested that microglia participate in cortical processing as a negative feedback mechanism like inhibitory neurons[84,85]. Our data raise an interesting possibility that region-specific increase of microglia density can act to regulate the influx of sensory signals from the thalamus to prevent over-excitation of the cortical circuit.

Furthermore, we identified spatiotemporal patterns of specific clusters of microglia in the corpus callosum and the cerebellar white matter[86]. These white matter-localized microglia may include the previously characterized microglial subset known as proliferative-region-associated microglia (PAM), which are situated in developing white matter tracts and regulate oligodendrocyte-mediated myelination into adulthood[87]. These microglia interact with neuronal, glial, and vascular cell types to orchestrate healthy brain development[88–92]. Previous studies showed that microglia might play key roles in shaping the development of white matter and survival of long-range projecting cortical excitatory neurons[86,87,93–95]. Rapid reductions of white matter-localized microglia and the expansion of gray matter microglia in telencephalic regions at P10 suggest that microglia may have two distinct roles in shaping the development of white and gray matter in the first and second postnatal week, respectively. The influence of microglia on cells in the local environment is limited by its vicinity with relatively short cellular processes. Hence, understanding the regional density of microglia and their changes across time can provide valuable insights into the extent of microglial influence on the development of individual brain areas[28,94].

The epDevAtlas is not without limitations. Our study focused on the crucial early postnatal period between P4 and P14. Prior research has indicated that glutamatergic neurons undergo programmed cell death before P4, influencing the subsequent development of GABAergic neurons[10]. To gain a better understanding of how excitatory and inhibitory balance is established in developing brains, future studies should explore earlier time points with glutamatergic cell types. To complement on this anatomical study, measuring the physiological and functional properties of individual GABAergic cells at various developmental stages could help estimate overall inhibitory tone in the brain and further explore the broader implications of these density changes in the development of functional networks. Additionally, while our chosen *Cx3cr1* mice offer relatively specific labeling of microglia, a minor population of the *Cx3cr1* gene is also expressed in other immune cells in the brain, such as border-associated macrophages[96,97]. Utilizing more specific transgenic reporters or combining them can greatly enhance the identification of microglia and their changes in developing mouse brains[92,98]. Further studies are also necessary to unravel the dynamic states, functions, and implications of microglia in the developing brain and their association with related diseases[99].

In summary, our growth charts represent a significant stride in comprehending crucial changes in cell types that are essential for typical brain development. This resource offers a systematic framework for evaluating pathological deviations across diverse neurodevelopmental disorders. Looking ahead, we envision employing epDevAtlas to include additional cell types, such as astrocytes, oligodendrocytes, and vascular cells in further investigations. This endeavor would produce more comprehensive and nuanced brain cell type growth charts, facilitating a deeper understanding of neurodevelopment.

## Methods

This study was conducted in accordance with all relevant ethical regulations. The study protocol was reviewed and approved by the Institutional Animal Care and Use Committee (IACUC) at Pennsylvania State University College of Medicine (PSUCOM).

### Animals

At Pennsylvania State University College of Medicine (PSUCOM), all experiments and techniques involving live animals conform to the regulatory standards set by the Institutional Animal Care and Use Committee (IACUC) at PSUCOM (protocol number: PROTO202001551). For labeling GABAergic cell types during early postnatal development (P4, P6, P8, P10, P12, P14), we crossed *Gad2*-IRES-Cre mice (JAX, stock 028867), *Sst*-IRES-Cre mice (JAX, stock 013044), or *Vip*-IRES-Cre mice (JAX, stock 031628) with Ai14 mice, which express a Cre-dependent tdTomato fluorescent reporter (JAX, stock 007908). Heterozygous *Cx3cr1*-eGFP[(+/−)] offspring for microglia analysis were produced by crossing homozygous *Cx3cr1*-eGFP mice (JAX, stock 005582) with C57Bl/6 J mice (JAX, stock 000664). These four animal lines were maintained and collected at PSUCOM.

Likewise, at the Allen Institute for Brain Science (referred to as the 'Allen Institute'), all animal experiments and techniques have been approved and conform to the regulatory standards set by the Institutional Animal Care and Use Committee (IACUC) at the Allen Institute (protocol number: 2105). For labeling cortical layer cell types at P6, P10, and P14, we used nine mouse genotypes. *Slc32a1*-IRES-Cre mice (JAX, stock 016962) were crossed with Ai65 reporter mice (JAX, stock 021875) and further crossed with *Lamp5*-P2A-FlpO mice (JAX, stock 037340) to produce triple transgenic offspring for layer 1 (L1) *Slc32a1* + /*Lamp5*+ cells. Layer 2/3 (L2/3) *Calb2*+ cells were labeled by crossing *Calb2*-IRES-Cre mice (JAX, stock 010774) with Ai14 reporter mice. Layer 4 (L4) *Nr5a1*+ cells were labeled by crossing *Nr5a1*-Cre mice (Mutant Mouse Resource & Research Center, stock 036471-UCD) with Ai14 reporter mice. For layer 5 (L5) *Rbp4*+ cells, *Rbp4*-Cre KL100 mice (Mutant Mouse Resource & Research Center, stock 037128-UCD) were crossed with Ai14, Ai193 (JAX, stock 034111), or Ai224 reporter mice (JAX, stock 037382). Layer 6 (L6) *Ntsr1*+ cells were labeled by crossing *Ntsr1*-Cre GN220 mice (Mutant Mouse Resource & Research Center, stock 030648-UCD) with Ai14 reporter mice. Layer 6b (L6b) *Cplx3*+ cells were labeled by crossing *Cplx3*-P2A-FlpO mice (JAX, stock 037338) with Ai193 or Ai227 (JAX, stock 037383) reporter mice.

Genotyping was performed by polymerase chain reaction (PCR) of tail biopsy genomic DNA for certain mouse lines. For mice younger than P6, *Rbm31*-based genotyping was used since visual identification of neonatal mouse sex based on anogenital distance is challenging[100]. All mice had access to food and water ad libitum, were maintained at 22 °C–25 °C with a 12 h light/12 h dark cycle, and both male and female mice were included in the study, with each animal used once for data generation. Detailed information on transgenic reporter lines and sample sizes per postnatal age and cell type analyzed in this study is available in Supplementary Tables 1–4.

## Brain collection, embedding, STPT imaging, and 3D reconstruction

The collection and STPT imaging of mouse brains have been extensively detailed in our protocol paper[101]. Briefly, animals were deeply anesthetized with a ketamine and xylazine mixture (100 mg/kg ketamine, 10 mg/kg xylazine, intraperitoneal injection) before perfusion. Transcardiac perfusion involved washing out blood with isotonic saline solution (0.9% NaCl) followed by tissue fixation with freshly made 4% PFA in phosphate buffer (0.1 M PB, pH 7.4). Post-fixation occurred by decapitating the heads and storing them in 4% PFA for 2 days at 4 °C. This was followed by careful brain dissection to ensure preservation of all structures. The brains were then stored in 0.05 M PB (pH 7.4) until STPT imaging preparation. Animals with incomplete perfusion or dissection were excluded from imaging and analysis.

At PSUCOM, precise STPT vibratome cutting was achieved by embedding fixed brains in 4% oxidized agarose in custom-built molds, ensuring consistent 3D orientation[101]. Cross-linking was achieved by incubating samples in 0.05 M sodium borohydride solution at 4 °C overnight before imaging. STPT imaging was performed using a TissueCyte 1000 (TissueVision) with a 20X objective lens (Olympus XLUMPLFLN20XW, 1.00 NA, 2.0 mm WD) and 910 nm two-photon laser excitation source (Chameleon Ultra II, Coherent). Green and red signals were simultaneously collected using a 560 nm dichroic mirror. Sampling rate (pixel size) was $1 \times 1 \mu m$ (xy) and image acquisition occurred at intervals of 50 $\mu m$ (z). We utilized custom-built algorithms to reconstruct STPT images into 3D volumes[52,101].

At the Allen Institute, fixed brain samples were embedded in 4% oxidized agarose and incubated overnight in acrylamide solution at 4 °C before heat-activated polymerization the following day (detailed protocol available at https://www.protocols.io/view/tissuecyte-specimen-embedding-acrylamide-coembeddi-8epv512njl1b/v4). Embedded samples were stored in 50 mM PB prior to STPT imaging using a TissueCyte 1000 with a 925 nm two-photon laser excitation source (Mai Tai DeepSee, Spectra-Physics). Green and red signals were simultaneously collected using a 560 nm dichroic mirror. Sampling rate (pixel size) was $0.875 \times 0.875 \mu m$ (xy) and image acquisition occurred at intervals of 50 $\mu m$ (z). Acquired images were transferred to the PSUCOM for further analysis.

## Reference brain template generation for epDevAtlas

Background channels of STPT-imaged data were used to construct morphologically averaged symmetric reference brain templates at ages P4, P6, P8, P10, P12, and P14. Templates were primarily generated using *Vip*-Cre;Ai14 mouse brains as the tdTomato signal from the fluorescent reporter is minimally visible once resampled to the template resolution of $20 \times 20 \times 50 \mu m$ (XYZ in the coronal plane). The P6 template was supplemented with data from *Gad2*-Cre;Ai14 and *Sst*-Cre;Ai14 data. The sample size per template varied between 6 and 14 brains from male and female mice.

To obtain symmetric templates, each preprocessed image underwent duplication and reflection across the sagittal midline. This step effectively doubled the number of input datasets used in the template construction pipeline, ensuring bilateral congruence. ANTs was utilized for registration-based methods to create a morphologically averaged symmetric template for each developmental age[76,77]. Morphologically averaged symmetric templates were created on Penn State's High-Performance Computing system (HPC) for each developmental age guided by the ANTs function, 'antsMultivariateTemplateConstruction2.sh' as described in Kronman et al.[40]. The procedure started by creating an initial template estimate from the average of input datasets. Following initialization, (1) Each input image was non-linearly registered to the current template estimate. (2) The non-linearly registered images were voxel-wise averaged. (3) The average transformation derived from registration was applied to the voxel-wise average image generated in the previous step, thereby updating the morphology of the current template estimate. The iterative process continued until the template's shape and intensity values reached a point of stability.

## Registration of CCFv3 labels to epDevAtlas templates

The P56 STPT Allen CCFv3 anatomical labels (RRID:SCR_020999) were iteratively down registered to each developmental timepoint represented by our STPT templates with the cell-type specific datasets acquired from *Gad2*-Cre;Ai14, *Sst*-Cre;Ai14, *Ntsr1*-Cre;Ai14, and *Rbp4*-Cre;Ai14 transgenic mouse lines, and the misaligned areas were corrected using manually drawn landmark registration[40]. The CCFv3 template was initially registered to the P14 STPT template utilizing the cell type-specific data and non-linear methods (Advanced Normalization Tools; ANTs)[76,77]. Subsequently, registration quality was assessed by superimposing the warped CCFv3 onto the P14 STPT template in ITK-SNAP[102] to identify critical misaligned landmark brain regions visually. After confirming the quality of registration, we applied the transformation derived from the registration to the CCFv3 anatomical labels, moving them to the P14 STPT template morphology. The misaligned labels were manually corrected using a 3D visualization and analysis software called DragonFly (Comet Technologies Canada Inc; Version 2024.1 for Windows; software available at https://dragonfly.comet.tech/). The results were discussed with another anatomist, and the labels were further corrected if needed. This process was repeated sequentially to align the anatomical labels from the P14 STPT template to the P12 STPT template, then to the P10 STPT template, and so forth until the P4 STPT template morphology was reached.

## Cell detection, image registration, and 3D cell quantification

We developed a flexible and automatic workflow with minimal annotations and algorithmic training based on our previous cell density mapping methods[31,39,52]. We employed ilastik as a versatile ML tool using random forest classification for signal detection[103], instead of more resource-intensive deep learning approaches that necessitate larger training sets and increased computational resources. Integrating ilastik into the automatic workflow, we designed our algorithms to perform parallel computations to detect each pixel with the maximum likelihood of it belonging to a cell, brain tissue, or empty space[33]. Detected signals that were deemed too small for cells were considered artifacts and discarded. Finally, we recorded the location of the center of mass (centroid) for each cell cluster. We performed image registration to map cell detection results to an age-matched epDevAtlas template using elastix[104]. The number of centroids was calculated for each brain region to generate 2D cell counting, which then was converted to 3D cell counting pre-established conversion factors (1.4 for cytoplasmic signals and 1.5 for nuclear signals)[31]. To calculate the anatomical volume from each sample, the epDevAtlas was first registered to individual samples using elastix and anatomical labels were transformed based on the registration parameters (see Supplementary Fig. 7). Then, the number of voxels associated with specific anatomical IDs was used to estimate the 3D volume of each anatomical area. 3D cell counting per anatomical regional volume ($mm^3$) was used to calculate the density.

To verify the cell detection algorithm's performance and accuracy, F-score analyses were performed across major time points (P4, P10, and P14), yielding an F-score of 0.96 for *Gad2*, 0.97 for *Sst*, 0.94 for *Vip*, and 0.97 for *Cx3cr1*, confirming the robustness of cell detection across developmental stages. Detailed validation results are provided in Supplementary Data 2. The cell counting algorithmic performance was evaluated using the F-score measure for true positives (TP), which represents the harmonic mean of precision (index for false positive (FP)) and recall (index for false negative (FN)), where 1 is the best and 0 is the worst. The F-score was calculated using the formula: $F-score = 2(precision \times recall)/(precision + recall)$
or $F-score = (TP)/[(TP+1)/2(FP+FN)]$.

## Multiplex FISH

To validate the expression profile of GABAergic neurons in the SSp, we utilized the RNAscope™ Multiplex Fluorescent V2 Assay (Advanced Cell Diagnostics, Hayward, CA, USA) for detecting *Gad2, Sst, Vip*, and tdTomato signals through ISH. Fresh frozen whole brain samples were collected from *Gad2*-Cre;Ai14, *Sst*-Cre;Ai14, and *Vip*-Cre;Ai14 mice at P4 and P10 (total $n = 24$; $n = 4$ per Cre-reporter line per age).

Briefly, following rapid cervical dislocation of mice, brains were immediately dissected and embedded in Optimal Cutting Temperature (OCT) media (Tissue-Tek, catalog #4853). The OCT-immersed brains were frozen using chilled isopentane and stored at −80 °C. Samples were sectioned in 10 µm-thick coronal brain slices at −20 °C using a cryostat and thaw-mounted onto slides before storage at −80 °C until needed. ISH was performed within 2 weeks of sectioning and according to the RNAscope™ Multiplex Fluorescent Kit v2 User Manual for Fresh Frozen Tissue (Advanced Cell Diagnostics, Inc., USA). Slides containing the specified coronal brain slices were fixed in 4% paraformaldehyde, dehydrated, hydrogen peroxide-treated for 10 min, and pretreated with protease IV solution for 30 min. During the initial hybridization steps, we applied the following probes from Advanced Cell Diagnostics: Mm-*Gad2* (Cat. #439371), Mm-*Sst*-O1 (Cat. #482691), Mm-*Vip* (Cat. #415961), and tdTomato-C3 (Cat. #317041), Mm-*Nkx2-1*-C2 (Cat. #434721), and Mm-*Prox1*-C2 (Cat. #488591). After probe hybridization, the slices underwent a series of probe signal amplification steps. Slides were counterstained with DAPI, and coverslips were mounted using Vectashield Hard Set mounting medium (Vector Laboratories).

Imaging was performed using a Zeiss LSM 900 with Airyscan 2 confocal microscope equipped with an automatic z-stage and Axiocam 503 mono camera (Zeiss, Gottingen, Germany). The acquired images were processed using Zen (3.1 blue edition and 3.0 SR black edition, Zeiss) software. For each Cre-reporter line, we examined mRNA-expressing cells in the SSp region from one hemisphere per brain section of P4 and P10 mice for colocalization with tdTomato mRNAs, specifically focusing on *Gad2, Sst*, or *Vip*. Neurons were identified based on the presence of a DAPI-stained nucleus and/or a distinct cell-like pattern of fluorescent mRNA dots and colocalization analysis was performed by two individuals in an unbiased, blinded method using the ImageJ Cell Counter plugin[105].

## Data visualization and isocortical flatmap generation

To visualize cell type density across different isocortical regions, we utilized MATLAB (MathWorks, version R2020a) and a custom MATLAB script to map Allen CCFv3 registered signals onto a 2D projected isocortical flatmap[52]. First, an isocortical flatmap was generated for individual sample datasets, using 3D counted cell data registered to the Allen CCFv3. For each postnatal timepoint per cell type, the flatmap images were averaged using a MATLAB script. Then, the averaged flatmaps were normalized for isocortical volume since the data were registered to Allen CCFv3 adult brain template. For image normalization, multiplication factors were determined for each postnatal timepoint (P4 to P14) by taking the average isocortical volume from age-matched developing brains and dividing those values by the average adult mouse isocortical volume. After normalization, each isocortical flatmap represented the average density of each cell type at a specific postnatal timepoint. To visualize cell type density across major regions of the entire brain, we created cell density maps using the bubble chart template in Excel (Microsoft, v.16.72). The size of each bubble corresponds to cell density values. Additionally, we plotted region-specific cell densities over time for each cell type using Prism (GraphPad, v.9.5.1).

## Statistical analyses

All data were presented as the mean ± standard deviation (SD). Significance was determined by a two-tailed p-value of less than 0.05. Prior to performing statistical analyses (GraphPad Prism, v.9.5.1), the datasets were assessed for normality and homogeneity of variance to check if the assumptions for parametric tests were met. Since the datasets did not meet these criteria, differences between groups were analyzed using Welch's one-way analysis of variance (ANOVA) followed by a non-parametric Dunnett's T3 post hoc multiple testing corrections test. Adjusted p-values from the multiple comparisons test were used to determine significance. Quantified cell type density data were collected, organized, and provided in the Source Data (Microsoft Excel, v.16.72).

Bayesian Multilevel Modeling for Density Magnitude Comparisons:

We conducted a Bayesian multilevel modeling analysis using the brms package in R[106] to assess age-related differences in cell densities of *Gad2, Vip*, and *Sst* neurons and *Cx3cr1* microglia in the isocortex. The model included a Gaussian family, with age as a fixed effect and random intercepts for both region and biological sample to account for spatial and biological variability. We used the default weakly informative priors provided by brms to stabilize estimates without overly constraining the model. Markov chain Monte Carlo (MCMC) sampling was performed with 2 chains, each with 4000 iterations (2000 warm-up iterations), using the cmdstanr[107] backend for efficiency. The control parameter adapt_delta was set to 0.99 to improve convergence. Convergence diagnostics, including R-hat values and trace plots, were examined to ensure adequate convergence of posterior samples. To test for significant differences in cell density patters across ages, we evaluated a global hypothesis that all age group means were equal, with 95% credible intervals (CIs) to summarizing the range of values for the combined age effects. For specific age-group comparisons, post hoc pairwise comparisons were conducted between age groups using estimated marginal means (emmeans)[108] to calculate 95% Highest Posterior Density (HPD) intervals for each pairwise contrast, indicating a credible difference. The model output included posterior mean estimates, full posterior distributions, and 95% credible intervals to comprehensively characterize the uncertainty in parameter estimates.

Functional Data Analysis for Assessment of Density Topographies:

To examine age-related spatial patterns of cell density in the isocortex, we applied functional data analysis (FDA) using a 3D tensor basis approach, leveraging the scikit-fda package in Python[109]. Density data for each cell type were modeled using a 3D tensor basis constructed from B-splines for each spatial dimension (x, y, z), where the xyz coordinates corresponded to the centroid coordinates of each region of interest (Supplementary Fig. 5). These coordinates were calculated from Allen CCF data using the Allen Software Development Kit, enabling accurate spatial representation within the native 3D brain structure. For each cell type, we computed coefficients for the tensor basis expansion by fitting a least-squares solution to density data across 37 cortical regions, generating a functional representation of the 3D density topography. To capture major spatial trends, we conducted functional principal component analysis (FPCA), selecting the optimal number of components based on cumulative variance explained (target: 99%). Outliers were identified by calculating L2-norm reconstruction errors, following scikit-fda's approach. We

projected density data into the reduced PC space, reconstructed it, and calculated deviations between original and reconstructed data. Using the 99th percentile of reconstruction errors as a threshold, we identified outliers for further examination by age group. Age-related differences in density topographies were evaluated through permutation ANOVA on mean reconstruction errors across ages (10,000 permutations). For significant ANOVA results, we performed pairwise permutation tests to identify specific age-group differences (Supplementary Fig. 6). False discovery rate (FDR) correction was applied to control for multiple comparisons, and p-values were adjusted using the Benjamini-Hochberg procedure to ensure robust pairwise significance estimates.

### Inclusion and Ethics Statement

All collaborators of this study who met the authorship criteria mandated by Nature Portfolio journals have been recognized as authors, as their involvement and contributions were crucial for study implementation. Collaborators agreed upon their respective roles and responsibilities ahead of the research.

### Reporting summary

Further information on research design is available in the Nature Portfolio Reporting Summary linked to this article.

## Data availability

The epDevAtlas is an openly accessible resource package including age-matched templates and anatomical labels which can be viewed and downloaded via https://kimlab.io/brain-map/epDevAtlas/. Full resolution and mapped cell type data can also be found at https://kimlab.io/brain-map/epDevAtlas/. The epDevAtlas and related cell type mapping data were deposited in a public data repository, Figshare, with https://doi.org/10.6084/m9.figshare.29910476. Additionally, the generated data from this study are provided in the Supplementary Information and Source Data files. All data can be used without restriction.

## Code availability

All available codes are deposited in a public data repository (https://github.com/yongsookimlab). Custom code for serial two photon tomography image stitching is available at https://github.com/KimLabResearch/TracibleTissueCyteStitching. 3D Cell Type Quantification Pipeline for the Whole Mouse Brain at https://github.com/yongsookimlab/Whole_Mouse_Brain_3D_Cell_Counting_Pipeline. Cortical flatmap at https://github.com/KimLabResearch/CorticalFlatMap. Elastix and ANTs registration parameters are included in Supplementary Figs. 7 and 8. All codes can be used without restriction.

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

## Acknowledgements

This work was supported by a grant from the National Institute of Health (NIH) grants RF1MH12460501, R01NS108407, and RM1MH138309 to Y.K. and T32NS115667 to D.P. Our deepest thanks to members of the BRAIN Initiative Cell Census Network for their insights. We express gratitude to all members of the Yongsoo Kim Lab for their motivation, commitment, and knowledge. We are utmost grateful toward all team members from the Allen Institute for Brain Science for graciously providing their time and efforts, including but not limited to animal husbandry, microscopy imaging (Nhan-Kiet Ngo and Nadezhda I. Dotson), and data management. We thank Julie Nyhus from the Allen Institute for managing all collaborative efforts during this project. We also thank Justin Silverman (Penn State Center for Neural Engineering) for advice on bioinformatic data analysis. Additionally, we thank BioRender.com for their illustration generation platform and the high-performance computing (HPC) center at PSUCOM.

## Author contributions

Corresponding author Y.K. conceived the project, supervised data generation and analysis, and edited the manuscript. First author J.K.L. collected data, performed data analysis, and wrote the manuscript with help from co-authors. S.B.M., D.S., Y.B., M.T., S.W., H.Z., B.T. and L.N. provided experimental support, generated data, and performed quality control. F.N.K. and H.P. helped to create epDevAtlas templates and labels. Y.W. developed the cell counting pipeline and D.P. assisted in conducting statistical analyses. D.J.V. and H.P. generated the web visualization platform.

## Competing interests

All authors declare no competing interests in both financial and non-financial interests.
