## [Transparent Peer Review file · Nature Communications]

epDevAtlas: Mapping GABAergic cells and microglia in the early postnatal mouse brain

Corresponding Author: Dr Yongsoo Kim

Version 0:

Reviewer comments:

Reviewer #2

(Remarks to the Author)

I now had time to review the manuscript that was re-submitted to Nature Communications. I had no specific comments that needed to be addressed in the previous version, and don't have new reservations with this version. The manuscript is still fit to be published and will be extremely valuable to a broad community.

(Remarks on code availability)

Reviewer #3

(Remarks to the Author)

The authors have fully addressed this reviewer's previous concerns. This reviewer has no further comments and endorses publication in its current form.

(Remarks on code availability)

Response to the decision on "epDevAtlas: Mapping GABAergic cells and microglia in the early postnatal mouse brain."

Editorial comments

Concerns about the robustness of the data

Our manuscript has two highlights: 1. Early postnatal 3D atlas templates with anatomical labels as a key resource for cell type mapping and other data integration in developing mouse brain. 2. Demonstrated use of the new atlas to quantitatively map spatiotemporal trajectories of GABAergic cell types and microglia in the whole mouse brain. All three reviewers agreed on the usefulness of our new atlas and praised the robustness of quantitative methods.

The reviewer 1 raised concern on the value of the data largely based on two specific reporter mice (SST- and VIP-Cre) for the programmed cell death in the cortex during early postnatal stage. However, the purpose of that specific section is to examine whether SST and VIP neurons have regionally distinct changes in their number/density to establish the mature circuit. Through additional experiments during our revision, we presented solid evidence that SST and VIP expression increases in post mitotic GABAergic cortical neurons during the early postnatal period and our chosen reporter lines faithfully recapitulate increased transcription activity of these key cell type markers. SST and VIP are key neuropeptide that will mark maturing interneurons successfully acquiring these cell type markers. Our mapping results identified several novel findings including specific cortical areas (e.g., medial prefrontal cortex) showing accelerated cell density decrease compared to other areas (e.g., somatosensory cortex) for SST neurons, providing evidence for regionally distinct cortical circuit formation during the early postnatal period.

Reviewer 1's main criticism is based on these two lines, not showing expected reduction of cell counting based on the program cell death. However, we would like to emphasize that our focus in this part is to quantify maturing interneurons with cell type markers in different cortical areas. Furthermore, we separately addressed the question of postnatal programmed cell death using Gad2-Cre in Figure 3. Hence, the data is robust in terms of methodology and support our biological conclusion.

Breadth of appeal of a cellular atlas focusing on GABAergic neurons and microglia in the P4-P14 mouse brain for our readership

There has been huge effort to examine cell type architecture in the brain, exemplified by the NIH BRAIN Initiative's effort. The current manuscript is a part of publication package with Nature publishing group (Handling editor: Noah Gray at Nature), in agreement with the wiring diagram working group in the BRAIN Initiative Cell Atlas Network (BICAN). This manuscript is heavily featured in the flagship paper, currently under preparation to be published in Nature, to summarize a series of working group manuscripts. Hence, the manuscript will be highly visible in broad neuroscience community via NIH BRAIN Initiative and many other related channels.

This project was strategically funded by the BRAIN Initiative to support on-going and future projects to comprehensively examine molecular and cell type architecture in the developing mouse brain. The current atlas is being used by BICAN groups as spatial framework to map spatial transcriptomic and other whole brain datasets. Similarly, our new atlas with ontologically consistent anatomical labels from Allen CCFv3 map will serve as key resource to interpret anatomical context of data with emerging techniques.

Through the revision, we provided additional evidence to support the validity of cell type specific reporter mice and the robustness of our approach. Our cell type map is the first of its kind to understand how our chosen key

cell types emerge in regionally distinctive patterns, which will serve as a standard chart to understand normal brain development and its deviation in animal models with pathological conditions. Indeed, we recently secured a NIH RM1 center grant (RM1MH138309) as a part of SSPsyGene consortium (<https://sspsygene.ucsc.edu>) to examine altered brain cell type growth chart from knockout mouse models with 100 different autism risk genes. Our data from current manuscript will powerfully advance our understanding of normal brain development and serve as a key data to understand brain changes in neurodevelopmental disorders. We also would like to emphasize that our microglia mapping data represent the first comprehensive quantitative mapping data in developing mouse brains, identifying a dramatic density shift from the white matter track to the grey matter between postnatal day (P) 8 and 10. This data will serve as foundational data for future mechanistic studies in the glia field to understand microglia roles in regionally distinct circuit maturation. Hence, we believe that our data has broad appeal to neuroscientists working on both neuronal and non-neuronal cell types.

Concerns on the quantification and classification of neurons (Reviewers 1 and 2), and the correlation between STPT templates and MRI-FA templates (Reviewer 3)

We will provide detailed response to individual reviewer comments below. Overall, the reviewer 1's main criticism centered on the choice of our reporter mice (SST and VIP) while reviewers 2 and 3 are highly supportive with very minor comments. We encourage the editorial team to put more weight on the overall value of main deliverables (new developmental atlas and whole brain cell type growth chart) of the manuscript rather than specific details raised by the reviewer 1 on the cortical interneuron program cell death.

Reviewer #1 (Comments for the Author):

This is the re-review of the manuscript from Liwang et al.

I am afraid the reviewers have not taken my comments on presenting cell numbers as a complement to the densities seriously. This means that conclusions about naturally occurring cell death have little validity.

We provide a full list of cell numbers across different brain areas per cell types and cortical layers counting in the Extended Data Table 3 and presented a few key cell numbers in main figures (e.g., Figures 3c and 4c, k). We performed F-score measurement to support cell counting results based on automatic detection methods. Moreover, fluorescence in situ hybridization experiment using RNA scope validated that our reporter (tdTomato) based detection faithfully represent endogenous gene expression patterns.

Figure 3 with Gad2-Cre;Ai14 reporter mice is directly related to the naturally occurring cell death. In the revision document, we clarified that we observed about 20% reduction from the peak Gad2+ cell number (~694K) at P10 to ~554K at P14, a little less than a previous study using Gad1 (or Gad67)-GFP reporter mice (PMID: 23041929). We explained that the difference is mainly that Gad2 is expressed in more mature neurons and appears later than Gad1 (PMID: 9831046). Hence, Gad2 expressing cortical interneurons were less likely to show naturally occurring cell death. We believe that these results do not affect the validity of our results. Rather, our data presents solid data on how much Gad2 expressing cortical interneurons undergo naturally occurring cell death, in comparison to the prior research using Gad1 (or Gad67)-GFP reporter mice (PMID: 23041929).

The density of interneurons as a measure is only biologically meaningful for neuronal computation in either comparison with the density of pyramidal cells or if the ratio between neurons/interstitial space/non-neuronal cells is known.

We strongly agree with the insightful comment. Indeed, this is the primary reason why we created new early postnatal atlas as spatial framework to enable such work. Here, we present our initial cell type mapping results

for GABAergic neurons and microglia as the first-of-its-kind in the early postnatal brain. Our 3D maps with image registration tools allowed us to precisely quantify volume (interstitial space) of individual areas in developing mouse brains, which we used to calculate the cell type density across more than 500 different brain areas in addition to the cell number. Hence, our data illustrate how these cells are populating different circuits with non-uniformly changing interstitial space. Surely, our atlas framework can be used to quantify different glutamatergic and other non-neuronal cell types in the future. However, performing additional cell type mapping will be beyond the scope of the current manuscript. Nevertheless, our work signifies the important initial step to examine cell type changes in postnatally developing brains while providing new atlases as a key data resource.

The authors also did not perform a detailed investigation of the increase of VIP/SST neurons. The fact that there is such a large increase in cells throughout development across reporters is damning to the validity of the method. The two sentences added in response to my 4th point drastically decrease the trust that we can put in the data from earlier time points.

We provided two additional evidence to confirm that early postnatal increase of VIP/SST is mainly from increased promoter activity using RNAscope RNA in situ hybridization and Allen gene expression database. We also confirmed that SST and VIP neurons are largely colocalized with MGE and CGE markers, respectively, further validating the faithful expression of our reporter mice. We performed F score calculation to provide quantitative evidence that we are accurately quantifying tdTomato labeled target cell types.

When we quantified VIP neurons, we refer to VIP expressing cell, not CGE derived immature neurons deemed to be VIP neurons. The same applies to the SST neurons, not MGE derived neurons deemed to be SST neurons. Both VIP and SST are key neuropeptides in mature interneuron subtypes that continue to increase its expression during early postnatal period (PMID: 27445703; PMID: 1972040; PMID: 8095906; PMID: 18598846; PMID: 32589877). Our data and RNA in situ data from Allen developmental gene expression database (Extended Date Figure 2) showed similar increased number of cortical neurons with these mature cell type markers. Hence, we believe that our quantification of these two cell types remain solid and well-justified in the context of tracking spatiotemporal trajectories of SST and VIP neurons, not in the context of naturally occurring cell death.

To clarify our intention for this part, we added the following text in the SST and VIP section.

“Sst and Vip cortical interneuron subtypes undergo differential growth patterns

Sst and Vip are key neuropeptides in maturing cortical interneuron subclasses that increase their expression during early postnatal period to support circuit maturation (PMID: 27445703; PMID: 1972040; PMID: 8095906; PMID: 18598846; PMID: 32589877). Disruption of Sst and Vip expression has been implicated in neurodevelopmental disorders (PMID: 33794534; PMID: 34848882). Here, we examine how Sst and Vip interneuron subclasses populate different cortical areas during the early postnatal period using Sst-Cre or Vip-Cre mice crossed with Ai14 reporter mice, respectively ^{11,56}.”

My other points have been adequately addressed, but I am afraid that the above flaws make it impossible for me to recommend publication in Nature Neuroscience.

Reviewer #2 (Comments for the Author):

The revised manuscript has addressed most of this reviewer's concerns, and the reviewer appreciates the thoughtful corrections made. A point for improvement is outlined below.

Minor concerns:

#1. (Fig.2e.g.i) Thank you for providing additional explanations for Figures 2e, g, and i. However, concern remains regarding the non-quantitative criteria used to categorize the groups. For example, the trends observed in AON and PR in Figure 2e appear to suggest a decreasing trend, which could potentially place them within the group defined in Figure 2i. While it is acknowledged that the scales differ notably, the lack of

quantitative criteria for grouping leaves the rationale unclear. If no quantitative metric is available, an explanation based on grouping by the ratio between the minimum and maximum values within a defined range would provide a more acceptable and comprehensible justification.

Thank you for the helpful feedback. Our grouping is based on anatomical division rather than the density change patterns. In the legend of Figure 2, we noted that "regions selected in Figures 2d–i represent those with the most notable density trends," which remains valid and supports our results. However, we recognize that this description may not fully clarify how these groups were chosen.

To provide more context, after mapping GABAergic neurons from P4 to P14 using STPT imaging and our cell quantification pipeline, we identified the highest Gad2 expression in specific brain regions, particularly the olfactory regions and striatum (Fig. 2b, c, d, f). Since these areas exhibited the highest densities of GABAergic neurons, we hypothesized that their subregions might follow similar density trends, as classified in the Allen CCFv3 brain structure taxonomy. Our analysis revealed distinct density trends within the olfactory system, with differences between the olfactory bulb regions (MOB, AOB) and the olfactory cortical regions (AON, PIR). This divergence, despite these subregions belonging to the same parent olfactory region, was highlighted in Figure 2e. Similarly, in the striatum, we observed a consistently increasing trend in both the CP (dorsal striatum) and ACB (ventral striatum). For the hippocampal formation (HPF), a region of the telencephalon, we hypothesized that Gad2 expression would align with that of the isocortex, which was confirmed upon comparing their densities (Fig. 2i). Therefore, our selection of these regions was largely based on their hierarchical anatomical organization with the observed Gad2 expression trends.

#2. (Fig.6h) The additional explanation provided for Figure 6h has improved clarity, particularly with the visualization of the data. However, the statistical analysis performed requires further elaboration. It is unclear what comparisons were made, what statistical methods were used, and whether multiple testing corrections were applied. Currently, the statement on P12, marked with an asterisk, appears to support the claim: "During the second postnatal week, we observed a further increase in CPvI microglial density compared to other CP subregions, which was significant at P12 (Fig. 6h)." However, there seems to be no statistical evidence provided to support the claim regarding the emergence of differences in CP subregional density over time: "This difference in CP subregional density was not evident at P4 and began to emerge at P6, aligning with our results showing that microglial density starts to prominently populate the SSp at this time (Fig. 6h)." If the emergence timing is critical to the claim, statistical analysis should be included to substantiate it. If such analysis is not feasible, the tone of the claim should be moderated accordingly.

We appreciate the reviewer's feedback and the opportunity to clarify our statistical analyses further. Dunnett's post hoc multiple comparisons test was performed after one-way ANOVA to account for potential differences in variances, specifically for **comparisons across caudoputamen (CP) subregions**. We analyzed the averaged microglial density data from all four CP subregions at each individual timepoint (e.g., P4, P6, P8, etc.) to determine whether differences in microglial density were present across subregions at a given developmental stage. After multiple testing corrections, significant differences were only observed between CP subregions at P12, which is now indicated on the graph using a black number sign or hash sign (#) instead of an asterisk.

Previously, we stated in the manuscript that "the difference in CP subregional density was not evident at P4 and began to emerge at P6,". To provide statistical support on the result, we performed multiple corrections testing to **compare averaged microglial densities across timepoints**. For each CP subregion, we analyzed density data across six developmental timepoints (P4 to P14) to assess the temporal dynamics of microglial density changes. Results showed no significant differences at P6; however, significant differences emerged at P8 for CPvm, CPdl, and CPvl, and at P12 for CPdm. To highlight these findings, color-coded asterisks corresponding to CP subregion-specific data points have been added to Figure 6h. Additional results from this analysis have been provided as an added spreadsheet in Extended Data Table 4.

The figure legend was updated to reflect the revision and the following text in the Methods section has been updated with the following: "differences between groups were analyzed using Welch's one-way analysis of variance (ANOVA) followed by a non-parametric Dunnett's T3 post hoc multiple testing corrections test. Adjusted p-values from the multiple comparisons test were used to determine significance."

The main result section has been updated with the following text.

“This difference in CP subregional density was not evident at P4 and began to significantly increase at P8, aligning with our results showing that microglial density starts to prominently populate the SSp at this time (Fig. 6g, h; Extended Data Table 4).”

#3. (Methods, 8. Statistical analyses) This reviewer appreciates updates to this section. The explanations regarding “Bayesian Multilevel Modeling for Density Magnitude Comparisons” and “Functional Data Analysis for the Assessment of Density Topographies” have been reviewed. While the intent is generally understood, these methodologies fall outside the reviewer’s area of expertise, and therefore cannot be evaluated in detail. Regarding Figures 3e, 4f, 4n, and 6f, the current versions appear to account for the effects of multiple comparisons. The previous figures, although lacking multiple testing corrections, were still considered reasonably valid representations, provided that remarks addressing the absence of such corrections were included. Including both the updated Bayesian-based figures and the previous versions as complementary representations would be beneficial. This approach would provide a more comprehensive perspective and support the robustness of the findings.

Thank you for your feedback on improving the statistical representations in the figures for better clarity and method comparisons. We have updated Figures 3e, 4f, 4n, and 6f to include the pairwise comparison plots from both the previous version (functional data analysis results) and the current version (Bayesian modeling), which accounts for multiple testing corrections.

Reviewer #3 (Comments for the Author):

This manuscript is a revised version of an original manuscript describing the EpDevAtlas and its applications to the generation of post-natal developmental maps of GABAergic neurons and microglial cells.

I would like to thank the authors for adding important validation experiments using in situ hybridization to characterize the expression coverage of the transgenic lines, and the addition of adult data points. The addition of microglial morphologies, asked by another reviewer is also an important improvement.

We appreciate your kind comments and helpful feedback, as the revisions greatly improved our study.

I am only puzzled by the new extended figure 1, showing a perfect correlation between STPT templates and MRI-FA templates at P4 and P14. I am surprised that the correlation coefficient is 1, and that the slope of the fit is very close to 1, showing that the histological procedure incurs absolutely zero deformation. I couldn't access the MRI files that are posted online for independent verification (due to a technical problem in the API of the hosting service). I would like to trust this impressive result, but would just ask the authors to carefully double check that this result is correct, and that no mistake occurred in this calculation and registration.

Thank you for your comment. To the best of our ability, we thoroughly double-checked all variables involved in the process of validating the STPT templates. These include the STPT and MRI-FA templates at both P4 and P14, as well as the registration data output, which includes specified brain region volume measurements for both the native STPT template and the registered STPT-to-MRI-FA image. We found minimal volume changes (about 5% at P4 and about 2% at P14) of our STPT based templates when compared to the MRI based templates. Correlation of individual regions between STPT and MRI template showed a correlation coefficient of 1 (or 0.9999), suggesting lack of non-linear volume changes with our sample preparation. The result is included in Extended Data Table 2, “MRI_STPT” tab. We updated our result section with the following text.

“Comparisons of our STPT-based templates with DevCCF magnetic resonance imaging (MRI)-based templates at P4 and P14, and with a previously published MRI study⁴⁷, demonstrated minimal (less than 5%) volume changes with almost no morphological deformations in our STPT- based templates (Extended Data Figure 1; Extended Data Table 2).”

A mention in the discussion that LSFM registration with tissue clearing was not tested would be useful to the reader, as this is something many groups will want to test.

The following statement has been included in the Discussion to offer more clarity: “Registration of images acquired by light sheet fluorescence microscopy (LSFM) with tissue clearing was not tested using the STPT templates in the current study. However, this is an area we are actively working on and plan to explore further in future research.”

If the authors are confident in the MRI registration and its impressive correlation with the STPT template, I consider this manuscript good for publication.

Thank you

Reviewer response

There is no reviewer comment to address in this resubmission